# A Survey of Hybrid Inference Systems for Large Language Models

## Abstract

Efficient deployment of large language models (LLMs) requires balancing inference speed with output quality. Speculative decoding accelerates inference by using a smaller draft model to propose future tokens, whereas reasoning-heavy approaches—such as chain-of-thought prompting, ensembles, and dynamic routing—improve output quality through deep search and verification. Although historically treated as isolated research trajectories, the demands of complex, high-entropy tasks have forced these domains to converge. This paper presents a structured taxonomy and analysis focused specifically on the emerging paradigm of hybrid and orchestrated inference systems. We begin by examining isolated approaches and their limitations, highlighting the necessity of this shift and identifying a critical "Orchestration Gap" in current architectures. We intend this work to serve as a catalyst for future research on orchestrated inference, ultimately contributing to the development of systems that are both fast and capable.

## 1 Introduction

The deployment of large language models (LLMs) has reshaped the field of artificial intelligence, shifting the focus from static classification tasks to dynamic, generative reasoning. While these improvements bring clear benefits, scaling these multi-billion-parameter architectures encounters a practical bottleneck: hardware constraints limit fast autoregressive inference, and algorithmic limitations demand longer, more deliberate reasoning to solve complex problems.

Addressing the latency constraint, a significant body of research has focused on inference acceleration, with speculative decoding emerging as one of the dominant paradigms (Chen et al., 2023a; Leviathan et al., 2023; Xia et al., 2024b). Speculative decoding accelerates LLM inference by using a smaller, computationally cheaper draft model for proposing candidate tokens, which are then verified by a larger, more expensive verifier model in a single forward pass. This "draft-then-verify" architecture reduces the effective cost of autoregressive decoding, making more efficient use of GPU memory bandwidth.

A second line of research has progressed separately, focusing on quality optimization through inference-time computation with methods such as Chain-of-Thought (CoT) (Wei et al., 2022) or Tree of Thoughts (ToT) (Yao et al., 2023). However, treating inference acceleration and deep reasoning as isolated tracks is no longer viable. With models tackling increasingly complex, high-entropy problems, these two domains inevitably overlap. Speculative methods degrade when task uncertainty requires deeper thought, while deep reasoning strategies incur prohibitive inference-time overhead that prevents real-time deployment. The fundamental challenge of modern LLM deployment is no longer simply making models faster or more capable reasoners, but dynamically balancing the two.

---

During the preparation of this manuscript, the authors used large language models (LLMs) such as Claude to assist with language polishing, phrasing, and readability. The authors carefully reviewed and edited all text, and take full responsibility for the final content. All core ideas, taxonomy, and conclusions are strictly human-sourced.

Although recent works have begun to explore this intersection, the resulting efforts remain fragmented. Despite these localized advances, a critical deficiency remains: the lack of a unified control mechanism that allows systems to switch between latency-optimized and reasoning-optimized modes in real time, which we term the "Orchestration Gap".

To formalize this evolving landscape, this paper presents a structured taxonomy focused specifically on hybrid and orchestrated inference systems. While previous surveys have extensively covered either speculative decoding or reasoning mechanisms independently, none provide a unified architectural framework for dynamically orchestrating computation across distinct System 1 and System 2 models at inference time.

To clarify our positioning, Table 1 contrasts our work with existing survey papers, highlighting the critical gap in hybrid inference analysis.

Table 1: Comparison of Existing Surveys vs. Our Work

| Survey | Primary Focus | Speculative Acceleration | Deep Search & Reasoning | Hardware / Serving Co-design | Hybrid / Orchestrated Arch. |
|---|---|---|---|---|---|
| (Xia et al., 2024b) | Latency (System 1) | ✓ | ✗ | ✓ | ✗ |
| (Zhang et al., 2024) | Latency (System 1) | ✓ | ✗ | ✗ | ✗ |
| (Ryu & Kim, 2024) | Latency (System 1) | ✓ | ✗ | ✗ | ✗ |
| (Miao et al., 2023) | Latency (System 1) | ✓ | ✗ | ✓ | ✗ |
| (Hu et al., 2025a) | Latency (System 1) | ✓ | ✗ | ✓ | ✗ |
| (Wang et al., 2025f) | Reasoning (System 2) | ✗ | ✓ | ✗ | ✗ |
| (Sui et al., 2025) | Reasoning (System 2) | ✗ | ✓ | ✓ | ✗ |
| (Li et al., 2025d) | Reasoning (System 2) | ✗ | ✓ | ✗ | ✗ |
| Ours (This Work) | Convergence | ✓ | ✓ | ✓ | ✓ |

The divergence in Table 1 reflects a structural split that persisted for several years. Latency optimization grew out of the systems and hardware community, with foundational speculative decoding work appearing at ICML 2023 (Leviathan et al., 2023) and in concurrent preprints (Chen et al., 2023a), where the primary concern was GPU memory bandwidth and the success metric was tokens per second. Reasoning optimization emerged from the NLP community, with Chain-of-Thought appearing at NeurIPS 2022 (Wei et al., 2022) and subsequent work published at ACL, EMNLP, and ICLR, where benchmark accuracy on complex tasks was the target and computational cost was secondary. Publishing in separate venues and optimizing for different objectives, the two communities had little incentive to engage with each other's constraints. Their convergence is recent, and is driven purely by the demands of production deployment, where both constraints apply simultaneously.

## 2 Background and Preliminaries

### 2.1 Autoregressive Memory Bottlenecks

Large language models generate text sequentially, forcing the hardware to wait for the completion of each step before initiating the next. On modern hardware, and at the small batch sizes typical of interactive deployment, this sequential process is memory-bound rather than compute-bound (Yan et al., 2024; Zhong et al., 2024). To understand this, we analyze arithmetic intensity ($I$), a concept formalized by the Roofline Performance Model (Williams et al., 2009), defined as the ratio of floating-point operations (FLOPs) to memory bytes accessed during a forward pass:

$$I = \frac{\text{FLOPs}}{\text{Bytes}} \tag{1}$$

Generating a single token requires repeatedly accessing model parameters from High-Bandwidth Memory (HBM) to perform only a small number of matrix-vector multiplications. The GPU compute units remain idle for the majority of the cycle, waiting for weights to arrive. Consequently, upgrading to a GPU with a higher peak TFLOPS yields negligible speedups for decoding; improvements require higher memory bandwidth or algorithmic changes that increase arithmetic intensity.

## 2.2 Speculative Decoding Fundamentals

Speculative decoding aims to overcome memory bandwidth constraints by introducing a "draft-and-verify" paradigm (Chen et al., 2023a; Leviathan et al., 2023). Instead of using the larger, computationally expensive target model to generate tokens autoregressively, a smaller draft model generates $K$ candidate tokens. These candidate tokens are then verified by the larger model in a single parallel forward pass, exploiting the massive arithmetic capacity of modern GPUs that remains underutilized during standard serial decoding.

**Speculative Decoding**

Figure 1: The Speculative Decoding workflow: A draft model proposes $K$ tokens which are verified in parallel by the target model.

The theoretical speedup of this system depends on the acceptance rate $\alpha$ and the cost ratio $c$, where $\alpha$ denotes the probability that a drafted token is accepted and $c$ represents the relative cost of a draft step compared to a verification step. Under simplifying assumptions, the expected speedup can be formulated as:

$$\mathbb{E}[\text{Speedup}(K, \alpha, c)] = \frac{1 - \alpha^{K+1}}{(1 - \alpha)(cK + 1)} \tag{2}$$

## 2.3 System 1 vs. System 2 Reasoning

In order to characterize the tradeoff between speed and reasoning depth, we rely on the dual process framework formalized for LLMs (Weston & Sukhbaatar, 2023; Yu et al., 2024). System 1 represents fixed computation, behaving as a forward-moving probabilistic system with a constant computational cost for each token, irrespective of the task's semantic complexity. In contrast, System 2 employs variable computation, shifting inference from static next-token prediction to a structured search procedure over a latent reasoning space, effectively trading higher latency for improved reasoning accuracy. From System 1 to System 2 (Li

et al., 2025d) grounds this transition technically, detailing how reasoning LLMs such as OpenAI o1/o3 and DeepSeek-R1 operationalize System 2 behavior through Monte Carlo Tree Search, reinforcement learning fine-tuning, and hierarchical planning frameworks.

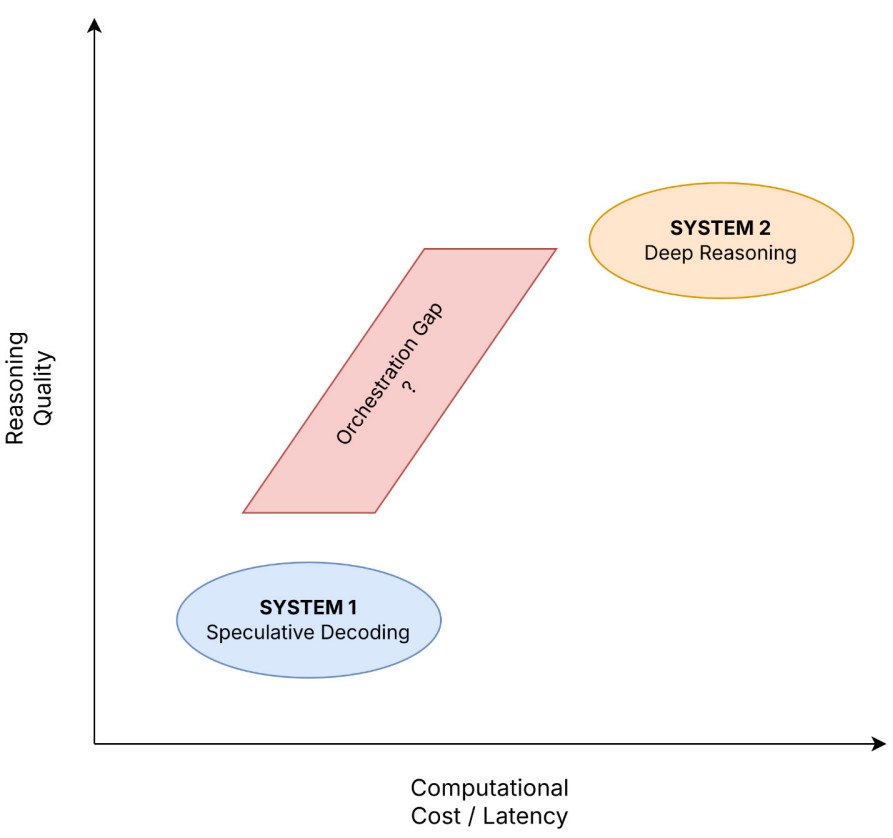

Figure 2: The Orchestration Gap: A visualization of the divergence between latency-optimized Speculative Decoding (System 1) and quality-optimized Deep Reasoning (System 2).

## 3 Independent Trajectories in Inference Optimization

Historically, efforts to optimize LLM inference have bifurcated into two independent trajectories: minimizing latency through memory bandwidth management (Latency Optimization), and maximizing reasoning depth through extended computation (Reasoning Optimization).

### 3.1 Latency Optimization

Modern inference-time latency optimizations—such as quantization, pruning, and architectural simplification—rely on lossy transformations, accepting some degradation in output quality in exchange for speed. In contrast, speculative decoding represents a prominent lossless acceleration paradigm (Chen et al., 2023a; Leviathan et al., 2023; Xia et al., 2024b). Rather than altering model weights or architecture, it restructures the decoding process itself through a draft–verify mechanism (see Section 2.2), enabling higher throughput while preserving the target model's output distribution.

Foundational approaches in this domain, such as Speculative Sampling (Chen et al., 2023a) and Fast Inference from Transformers (Leviathan et al., 2023), establish baseline efficiency gains using fixed speculation

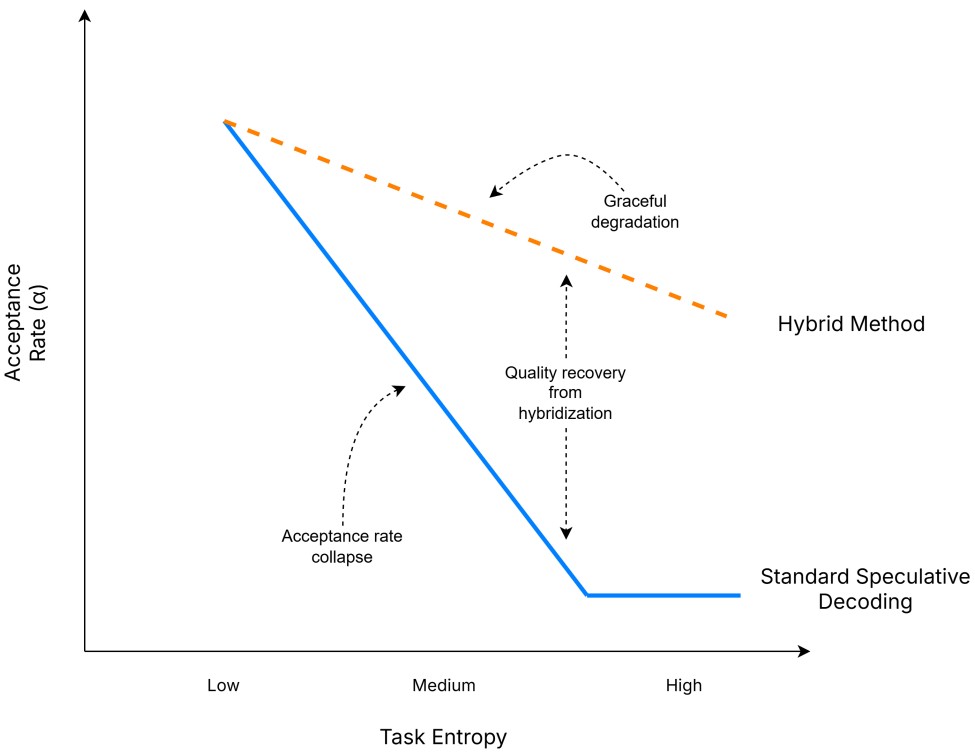

Figure 3: Acceptance rate ($\alpha$) vs. task entropy for standard speculative decoding and a hybrid method. While standard decoding suffers an acceptance-rate collapse at high entropy, the hybrid method achieves graceful degradation by maintaining substantially higher acceptance rates through hybridization. Curves are conceptual illustrations of qualitative trends and do not represent empirical measurements.

windows. However, because these static policies can waste compute during high-entropy phases, adaptive control frameworks like SpecDec++ (Huang et al., 2024) and DSDE (Yang et al., 2025a) emerged to dynamically adjust the speculation length based on real-time signals like KL divergence. Pushing this efficiency further, structural speculation methods such as SpecInfer (Miao et al., 2024) and EAGLE (Li et al., 2024c) transition away from sequential drafting altogether. By using tree- and graph-based structures to generate candidate tokens in a nonlinear fashion, these methods allow the verifier to evaluate and accept multiple tokens across different branches simultaneously, significantly increasing acceptance rates compared with those of a single linear chain. This paradigm has been rapidly advanced by subsequent iterations like EAGLE-2 (Li et al., 2024d), which introduces dynamic draft trees driven by real-time confidence scores, and EAGLE-3 (Li et al., 2025c), which abandons feature prediction entirely in favor of direct token prediction and introduces multi-layer feature fusion via a training-time test technique, achieving up to 6.5× speedup — approximately 1.4× over EAGLE-2.

To execute these complex topologies without bottlenecking memory bandwidth, hardware-aware kernels such as DeFT (Yao et al., 2025) have emerged to optimize tree-structured attention, utilizing KV-guided grouping to eliminate redundant memory loads when verifying divergent branches. Extending this line of work, DFlash (Chen et al., 2026) replaces autoregressive draft generation entirely with a lightweight block diffusion model that generates all draft tokens in a single parallel forward pass conditioned on target-model context features, achieving over 6× lossless acceleration and up to 2.5× higher speedup than EAGLE-3 by completely bypassing the sequential bottleneck of traditional draft models.

Rather than optimizing the interaction between two distinct models, subsequent research explored eliminating the traditional standalone draft model altogether to reduce VRAM overhead and inter-model coordination costs. This single-model paradigm has evolved through two distinct architectures. The first is single-model multi-head parallel prediction, as seen in Medusa (Cai et al., 2024), which adds multiple independent feedforward decoding heads on top of a frozen target-model backbone to predict several future tokens simultaneously. The second is self-speculative decoding, exemplified by LayerSkip (Elhoushi et al., 2024) and SWIFT (Xia et al., 2024a), which use a subset of the target model's own layers—exiting the model early to generate drafts, and using the remaining layers for verification.

At the extreme end of this efficiency spectrum, heuristic and training-free drafting strategies replace neural drafting entirely with lightweight mechanisms like prompt lookup, pattern matching, or retrieval (e.g., Prompt Lookup Decoding (Saxena, 2023) and DRAGIN (Su et al., 2024)). While this results in lower memory overhead and easier deployment, these approaches are highly task-dependent, causing acceptance rates to plummet under complex, high-entropy generation.

Despite their ability to accelerate inference, pure latency optimizations are inherently vulnerable to high-entropy tasks. Because efficiency relies heavily on the draft model accurately predicting the target distribution, complex reasoning or coding tasks degrade this alignment and cause acceptance rates to collapse. Consequently, the system loses its theoretical speedup and instead incurs a computational penalty, leading to negative acceleration. Furthermore, standard verification relies strictly on static probability matching rather than semantic correctness, meaning these methods optimize purely for speed without improving reasoning quality. This rigidity highlights the need for architectures capable of embedding quality constraints directly into the acceleration pipeline.

## 3.2 Reasoning Optimization

In parallel with latency optimizations, a separate research trajectory prioritizes generation quality over latency reduction, leveraging inference-time computation through deep reasoning search, adaptive routing, and multi-model verification.

Chain-of-Thought prompting (CoT) (Wei et al., 2022) introduced the paradigm of intermediate reasoning, demonstrating that explicitly generating a sequence of logical steps before producing the final answer significantly improves performance on complex tasks. Methods like Tree of Thoughts (ToT) (Yao et al., 2023) and Self-Consistency (Wang et al., 2023) extend this approach by replacing naive greedy decoding with non-linear branching and the sampling of multiple reasoning paths. Recently, internalized reinforcement learning frameworks like DeepSeek-R1 (DeepSeek-AI, 2025) have further advanced this paradigm, effectively shifting inference from static token prediction to a structured, self-guided search procedure.

Beyond structural search, contrastive and ensemble methods emerged to increase reliability by leveraging multi-model distributions. Ensemble approaches, such as LLM-Blender (Jiang & Lin, 2023) aggregate outputs from an ensemble of diverse models to cross-check and fuse a single high-quality response. Conversely, contrastive methods such as Contrastive Decoding (Li et al., 2023) use a smaller model as a negative baseline, subtracting its probability distribution from that of the larger target model to actively suppress hallucinations and provide a more accurate output.

To expand model capabilities further without proportionally increasing per-token compute, research also introduced static routing and tool-use frameworks. These methods operate at varying levels of granularity, from as micro-routing tokens to sparse experts (e.g., Switch Transformer (Fedus et al., 2021) and Mixtral (Jiang et al., 2024)), to dispatching complex sub-tasks directly to external APIs (e.g., Toolformer (Schick et al., 2023) and Gorilla (Patil et al., 2023)).

Despite their profound reasoning capabilities, these pure quality optimizations suffer from a critical limitation: prohibitive inference-time latency. Deep search frameworks such as ToT scale poorly for real-time applications, as the computational overhead grows exponentially in the worst case ($O(B^d)$) with increasing tree depth. Similarly, static ensemble methods demand linear scaling of verification costs ($O(M)$) (where M is the number of models) and suffer from the heavy computational overhead of invoking multiple models simultaneously, preventing scalable deployment. Furthermore, a shared limitation with isolated routing and

Table 2: Latency Optimization Methods

| Method | Drafting Paradigm | Target Model | Baseline Type | Speedup | Speedup vs. Standard SD |
|---|---|---|---|---|---|
| Speculative Sampling (Chen et al., 2023a) | Sequential linear drafting | Chinchilla 70B | Vanilla AR | 2.0–2.5× | Baseline (is standard SD) |
| Fast Inference from Transformers (Leviathan et al., 2023) | Sequential linear drafting | T5-XXL / LaMDA 137B | Vanilla AR | 2.0–3.0× | Baseline (is standard SD) |
| SpecDec++ (Huang et al., 2024) | Adaptive-length sequential drafting | LLaMA-2-Chat-70B | Vanilla AR[†] | 2.04–2.26× | 7.2%–11.1% over fixed-window SD |
| DSDE (Yang et al., 2025a) | KLD-adaptive sequential drafting | LLaMA-3.1-70B-Instruct | Vanilla AR[†] | 2.00–2.75× | Competitive with profiled static-opt SD |
| SpecInfer (Miao et al., 2024) | Tree-based multi-draft | OPT-30B / LLaMA-7B / LLaMA-65B | Vanilla AR | 1.5–2.8× (distributed); 2.6–3.5× (offloading) | 1.3–1.8× over seq. SD |
| EAGLE (Li et al., 2024c) | Feature-level autoregressive drafting | Vicuna-7B/13B/33B | Vanilla AR | 3.0× | ∼2× over standard SD |
| EAGLE-2 (Li et al., 2024d) | Dynamic draft tree | Vicuna-7B/13B, LLaMA-3-8B | Vanilla AR | 3.5× | ∼1.2× over EAGLE |
| EAGLE-3 (Li et al., 2025c) | Multi-layer fusion with direct token prediction | Vicuna-13B / LLaMA-3.3-70B / DSR1-LLaMA-8B | Vanilla AR | Up to 6.5× | ∼1.4× over EAGLE-2 |
| Medusa (Cai et al., 2024) | Multi-head parallel prediction | Vicuna-7B/13B/33B | Vanilla AR | 2.2–3.6× | N/R vs. standard SD directly |
| LayerSkip (Elhoushi et al., 2024) | Self-speculative early exit | LLaMA-2-7B/13B | Vanilla AR | 1.34–2.16× | N/R vs. standard SD directly |
| DFlash (Chen et al., 2026) | Parallel block diffusion drafting | Qwen3-8B / LLaMA-3.1-8B | Vanilla AR | Over 6× | Up to 2.5× over EAGLE-3 |

[†]SpecDec++ and DSDE report wall-clock speedups relative to a vanilla autoregressive baseline but are primarily designed to improve upon fixed-window speculative decoding. SpecDec++ measures improvement over a best fixed-$K$ SD baseline on the same model pair; DSDE measures against a profiled static-optimal SL baseline. Both also report vanilla AR speedups for comparability.

*Notes:* The Baseline Type column indicates the primary comparison standard used in each paper. "Vanilla AR" denotes a standard autoregressive decoding baseline without speculative acceleration. Methods marked with [†] additionally report their primary gains relative to a fixed-window or static-optimal speculative decoding baseline. Prompt Lookup Decoding (Saxena, 2023) and DRAGIN (Su et al., 2024) are omitted because their heuristic drafting mechanisms are highly task-dependent and their reported gains are not measured against a consistent baseline. DeFT (Yao et al., 2025) is excluded because it constitutes a kernel-level optimization rather than a standalone decoding method and thus does not yield an independently attributable speedup ratio.

tool-use approaches is their rigidity; once an expert is selected, they rely on static task-allocation and lack the dynamic capability to switch control mid-generation. The result is a static computational penalty on every prompt — one that makes these approaches impractical for real-time interactive deployment regardless of task complexity.

# 4 Hybrid and Orchestrated Architectures

In the context of this survey, we define a hybrid and orchestrated architecture as any system that dynamically manages the tradeoff between fast, inexpensive computation (System 1) and slow, deep computation (System 2) (Weston & Sukhbaatar, 2023; Yu et al., 2024). This hybridization is not limited to token-level modifications; rather, it spans multiple levels of the inference stack, balancing theoretical capability with operational reality.

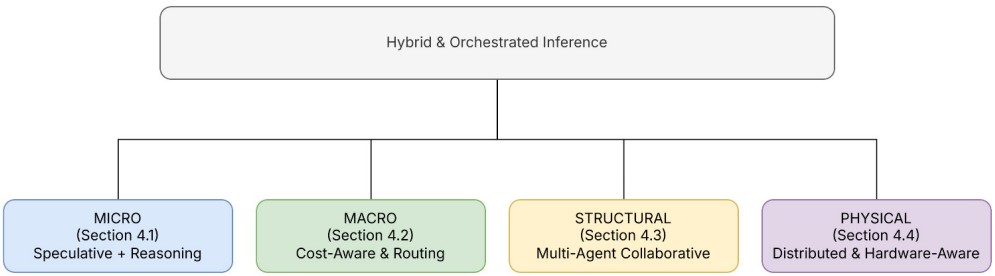

Figure 4: A taxonomy of Hybrid and Orchestrated Inference architectures, organized across four levels of granularity: Micro (Speculative + Reasoning), Macro (Cost-Aware & Routing), Structural (Multi-Agent Collaborative), and Physical (Distributed & Hardware-Aware).

To systematically analyze these efforts, we categorize hybrid architectures into four distinct levels of granularity:

- **Speculative + Reasoning Hybrids (Micro-Level):** Systems that balance speed and quality at the token level, integrating speculative acceleration with quality-aware filtering within the generation loop.

- **Cost-Aware & Routing Hybrids (Macro-Level):** Frameworks that balance speed and quality at the prompt level, dynamically allocating entire queries between cheap, fast models and expensive, deep reasoners.

- **Multi-Agent Collaborative Hybrids (Structural-Level):** Architectures that structurally merge the two domains, utilizing lightweight speculative drafters to feed heavy, collaborative expert ensembles.

- **Distributed & Hardware-Aware Hybrids (Physical-Level):** Systems where the orchestration between latency and reasoning is dynamically forced by real-time hardware constraints, such as memory bandwidth and compute capacity.

Together, these approaches demonstrate a multi-layered attempt to bridge the Orchestration Gap, though they often rely on fragmented or static heuristics rather than a fully unified controller. While we have placed each surveyed paper into the category that best represents its primary architectural contribution, several frameworks inherently span multiple levels of this taxonomy. We detail these papers within their primary categories and explicitly discuss these cross-boundary approaches later in Section 4.5.

## 4.1 Speculative + Reasoning Hybrids (Micro-Level)

The most granular approach to hybridization occurs at the micro-level, where systems balance speed and quality directly at the token or step level within the generation loop. While traditional speculative decoding accelerates inference, its verification mechanism relies strictly on static probability distribution matching. This algorithmic rigidity causes standard speculation to reject high-quality, semantically accurate tokens

simply because they diverge from the target model's exact vocabulary preferences. Micro-level hybrids solve this by embedding semantic, reasoning, and reward-aware constraints directly into the draft-and-verify pipeline, decoupling verification correctness from surface-level vocabulary overlap.

To bypass strict probability matching, recent architectures evaluate the semantic validity of drafted sequences. Judge Decoding (Bachmann et al., 2024) trains a compact judge module on top of the target model's hidden embeddings to evaluate the semantic correctness of drafted tokens, accepting misaligned but high-quality continuations that standard speculation would discard. Directly addressing the high training complexity of this approach, SelfJudge (Yoon et al., 2025) eliminates reliance on human-annotated ground truth by generating its own training supervision. By deriving a semantic preservation score entirely from the target model's likelihood differences, it enables automatic judge verifier training across diverse domains. Similarly, Beyond Tokens (Dong et al., 2026) probes internal transformer hidden state activations to verify sequence meanings, enabling the acceptance of lexically divergent but semantically equivalent drafts. Extending this logic, Think Before You Accept (Wang et al., 2025d) fuses standard next-token logits with reflective reasoning signals derived from the target model's own self-evaluation capacity, embedding a lightweight System 2 verification layer directly into the rapid generation loop. Other frameworks refine this filtering through continuous syntactic and semantic coherence checks (He et al., 2025), or by reformulating the acceptance objective entirely. MARS (Song et al., 2026b), for instance, replaces the standard criterion with a training-free, margin-aware verification strategy that conditions token acceptance on decision stability derived from the target model's output logits. It relaxes strict rejection only in low-margin regimes where the target model exhibits weak preference among top candidates, and it scales robustly across model sizes from 8B to 235B parameters while preserving generation quality.

Moving beyond semantic equivalence, token verification can also be governed directly by explicit human preference and reward signals. Reward-Guided Speculative Decoding (RSD) (Liao et al., 2025) integrates a Process Reward Model (PRM) directly into the verification step, dynamically accepting or rejecting drafted tokens based on reasoning quality and forcing recomputation when sequences are logically flawed. This alignment constraint is mirrored in Reward-Shifted Speculative Sampling (RSSS) (Li et al., 2025a), which continuously aligns drafted tokens with preference signals at inference time, and Guided Speculative Inference (GSI) (Geuter et al., 2025), which embeds reward-guided constraints to approximate optimal soft best-of-n alignment at a fraction of the latency cost. At the distributional level, Speculative Contrastive Decoding (SCD) (Yuan et al., 2024) dual-purposes the draft model: it proposes candidate tokens as usual, but its output distribution is simultaneously subtracted from the target model's distribution during verification, suppressing low-quality generation patterns without additional model overhead.

As models increasingly rely on intermediate logic, hybridization has scaled from token-level speculation to step-level reasoning. Lookahead Reasoning (Fu et al., 2025b) introduces a layer of parallelism where a draft model proposes entire reasoning steps, which the target model then verifies based on semantic equivalence. SpecReason (Pan et al., 2025) applies a similar logic, accelerating inference by speculatively executing intermediate "thinking tokens" with a small model and skipping expensive target model calls for straightforward logical steps. Expanding this into multi-path exploration, SpecCoT (Wang et al., 2025a) evaluates multiple chain-of-thought drafts in parallel using dynamic large-model fallbacks, achieving up to 4.1× speedups, while ThoughtMani (Liu et al., 2025) dynamically injects external chain-of-thought sequences generated by a smaller model inline between the thinking tokens of the base reasoning model. To improve the accuracy of these long-horizon drafts, ConFu (Qin et al., 2026) conditions the draft model on future-oriented "contemplate tokens" and soft prompts derived from the target model's forward pass, enabling it to anticipate complex reasoning trajectories.

Finally, hybridization occurs through dynamic token-level routing and real-time handoffs mid-generation. CITER (Zheng et al., 2025) trains a reinforcement-learning-based router on the base model's hidden states to dynamically switch each token between a small and large model at each step directly within the generation loop based on real-time difficulty. RelayGen (Song et al., 2026a) employs repeated bidirectional, segment-level switching mid-generation, delegating low-difficulty segments to a smaller model and returning control to the large model at subsequent high-difficulty segments, without requiring re-prefilling at each switch.

Conversely, Speculative Thinking (Yang et al., 2025b) tracks structural linguistic delimiters and reflective tokens (e.g., "wait") to trigger instantaneous upward handoffs, escalating to the large model precisely at the boundaries of deep reasoning segments. To manage uncertainty during these handoffs, Entropy-Aware Speculative Decoding (EASD) (Su et al., 2025b) embeds a dynamic statistical penalty into the verification step to automatically intercept and resample low-confidence reasoning paths. Pushing the boundaries of this micro-level integration, DREAM (Hu et al., 2025b) extends speculative hybridization to the multimodal domain, utilizing cross-attention target feature injection and an entropy-adaptive fusion mechanism to verify visually-grounded drafts inside the generation loop.

**Synthesis.** These micro-level hybrids represent a crucial first step toward fully unified inference. By embedding reasoning models, reward signals, and semantic verification directly into the acceleration pipeline, they enable quality-aware filtering at the token level. However, operating strictly within a predefined generation loop limits their scope, and several hard architectural ceilings remain:

- **Heuristic Coupling:** The coordination logic in these systems is largely static, relying on fixed confidence thresholds, hand-coded linguistic delimiters, or pre-tuned reward boundaries. Each intervention responds to local token-level signals rather than the overall reasoning difficulty of the task, with no global signal to reflect whether the generation as a whole is on track.

- **Fixed Model Horizon:** Micro-level hybrids are bounded by their model pair. When task complexity exceeds what the target model can handle, no amount of token-level steering compensates, and there is no mechanism to escalate to a more capable reasoner.

- **Auxiliary Model Overhead:** Methods that rely on separate judge modules, process reward models, or RL-trained routers reintroduce the deployment complexity that speculative decoding was designed to avoid. Keeping these components aligned and in memory alongside the draft and target models adds substantial VRAM overhead. Self-supervised approaches like SelfJudge (Yoon et al., 2025) partially mitigate this, but remain the exception.

This fixed model horizon directly motivates the macro-level routing paradigm explored in Section 4.2, where compute allocation is governed by query difficulty rather than token-level signals.

## 4.2 Cost-Aware & Routing Hybrids (Macro-Level)

Moving beyond token-level modifications, hybridization can also be organized at the macro level. This category assumes heterogeneity in query difficulty, operating on the premise that not all queries require the intelligence and computational cost of the most capable model. Rather than altering the internal decoding loop, these systems dynamically route entire prompts to the most appropriate model based on task difficulty. This approach balances System 1 efficiency with System 2 reasoning capacity on the fly, optimizing the trade-off between expected cost, system latency, and output quality. Dynamic Model Routing and Cascading (Moslem & Kelleher, 2026) provide a comprehensive characterization of this design space, categorizing multi-LLM routing systems along three axes: when routing decisions are made, what information is used to make them, and how they are computed—ranging from human preference signals to uncertainty quantification and reinforcement learning.

The initial wave of macro-routing research focuses on predictive routing via external classifiers. A foundational approach in this space is FrugalGPT (Chen et al., 2023b), which learns which combination of LLMs to invoke for a given query, calling cheaper models first and escalating to more expensive ones only when a trained quality check is not satisfied. RouteLLM (Ong et al., 2025) extends this paradigm by shifting from sequential cascading to predictive routing: it trains a lightweight classifier on preference data to predict which model can successfully answer a prompt at the lowest cost before generation begins. Similarly, Hybrid LLM (Ding et al., 2024) trains a binary classifier on query difficulty with a tunable quality threshold at test time, allowing practitioners to adjust the cost-quality tradeoff without retraining. Shifting these decisions to a pre-inference tuning stage, FORC (Šakota et al., 2024) frames model selection as an integer linear programming problem to optimize the global cost-performance tradeoff in batch. Frameworks like Routing to the Expert

Table 3: Micro-Level Hybrid Inference Methods

| Method | Hybrid Type | Target Model | Baseline Type | Speedup vs. AR Baseline | Speedup vs. Standard SD | Quality Metric |
|---|---|---|---|---|---|---|
| Judge Decoding (Bachmann et al., 2024) | Compact Embedding-Head Verification | LLaMA-3.1-8B / 405B | Vanilla AR | 9.0× | Superior to standard SD | Quality preserved on MT-Bench, GSM8K, ARC, MMLU; conditional on training domain |
| Beyond Tokens (Dong et al., 2026) | Internal-State Semantic Verification | DeepSeekR1-32B / QwQ-32B | Vanilla AR | 2.7× / 2.1× | Outperforms token-level and sequence-level SD baselines | Accuracy preserved on MATH-500, GPQA-Diamond |
| MARS (Song et al., 2026b) | Margin-Aware Verification | LLaMA / Qwen families (8B–235B) | Vanilla AR | Up to 4.76× | Superior to standard SD across all scales | Quality preserved across MT-Bench, HumanEval, GSM8K |
| Think Before You Accept (Wang et al., 2025d) | Reflective Logit Fusion Verification | LLaMA-3 1B / 8B / 70B | Standard SD | N/R vs. AR directly | Outperforms standard SD in acceptance rate | Accuracy preserved across prompt variations |
| RSD (Liao et al., 2025) | Process Reward Verification | Qwen-2.5 / LLaMA-3 | FLOPs vs. target-only† | Up to 4.4× fewer FLOPs vs. target-only | Superior to standard SD | Up to +3.5% accuracy on Olympiad-Bench, MATH-500, GSM8K |
| SpecReason (Pan et al., 2025) | Step-Level Speculative Reasoning | QwQ-32B + R1-1.5B | Vanilla AR | 1.4×–3.0× | Additional 8.8%–58.0% latency reduction over SD alone | +0.4%–9.0% accuracy vs. base model on AIME, MATH-500, GPQA |
| Lookahead Reasoning (Fu et al., 2025b) | Step-Level Semantic Verification | DeepSeekR1-Distill-32B + 1.5B | Vanilla AR | 1.04×–1.71× standalone; 2.11× combined with token-level SD | Lifts token SD from 1.4× to 2.11× | Accuracy within 1.0%–2.1% of AR baseline on GSM8K, AIME, GPQA |

†RSD reports FLOPs reduction rather than wall-clock speedup.

*Notes:* The Baseline Type column indicates the primary comparison standard used in each paper. "Vanilla AR" denotes a standard autoregressive decoding baseline without speculative acceleration. "Standard SD" denotes a conventional speculative decoding baseline with static probability-matching verification. Methods whose primary metric is not wall-clock speedup against a vanilla AR baseline are annotated accordingly. CITER (Zheng et al., 2025), EASD (Su et al., 2025b), DREAM (Hu et al., 2025b), SelfJudge (Yoon et al., 2025), and Speculative Thinking (Yang et al., 2025b) are omitted because their reported gains are measured against task-specific or model-specific baselines rather than a vanilla autoregressive baseline. SSS (Li et al., 2025a), GSI (Geuter et al., 2025), and SCD (Yuan et al., 2024) are excluded because they report accuracy improvements or FLOPs reductions without corresponding wall-clock speedup figures. SpecCoT (Wang et al., 2025a), ThoughtMani (Liu et al., 2025), and ConFu (Qin et al., 2026) are similarly omitted because they report only qualitative or relative improvements without standardized speedup ratios against a consistent baseline.

(Lu et al., 2024) and Expert Router (Pichlmeier et al., 2024) take a domain-centric approach, classifying incoming prompts to direct them to specialized expert models through explicit labeling and prompt-level clustering respectively. Large Language Model Routing with Benchmark Datasets (Shnitzer et al., 2023) empirically validate this paradigm, showing that learned routing strategies consistently outperform random or static model selection.

To eliminate the overhead of training external classifiers, a subsequent line of research shifts to training-free routing via internal confidence signals. AutoMix (Aggarwal et al., 2024) employs few-shot self-verification: a smaller model scores the reliability of its own answer, and a meta-verifier uses this score to decide whether to escalate to a larger model, thereby reducing costs without requiring an external routing head. LLM Cascades with Mixture of Thoughts (Yue et al., 2024) takes a different approach to the same problem, generating multiple reasoning representations of the same query—such as Chain-of-Thought and Program-of-Thought—and using their consistency as a training-free escalation signal without any external scoring model.

Rather than relying on trained proxy models or heuristic rules, recent frameworks mathematically optimize the entire model selection pipeline. Cascade Routing (Dekoninck et al., 2025) formally derives the optimality conditions for both routing and cascading, showing that a unified framework combining the two consistently outperforms either applied in isolation. MetaLLM (Nguyen et al., 2024) extends this to multi-model environments, employing a multi-armed bandit framework to dynamically assign queries across diverse LLM pools while adapting to defined cost budgets.

Translating these dynamic optimization models into practice, current research adapts routing mechanisms for the strict constraints of live deployment. MixLLM (Wang et al., 2025b) advances the multi-armed approach by employing a contextual bandit framework that jointly optimizes quality, API cost, and real-time system latency, actively preventing the router from overloading a single fast model during high query volume. To further refine these real-time decisions, recent work incorporates rigorous uncertainty quantification. Leveraging uncertainty estimation for Efficient LLM routing (Zhang et al., 2025) uses semantic entropy as a lightweight, training-free signal to guide model selection. Most explicitly bridging the System 1 and System 2 divide, CP-Router (Su et al., 2025a) uses a rigorous statistical framework to quantify uncertainty, ensuring that the transition to expensive reasoning models is only triggered when a smaller model's failure is mathematically likely.

**Synthesis.** Macro-level routing frameworks successfully optimize inference costs by explicitly measuring query difficulty before generation begins. By directing only the most complex prompts to resource-intensive System 2 models, these systems reduce average overhead without sacrificing task quality. Yet, these efficiency gains depend on a rigid architectural assumption that the initial routing decision is both flawless and permanent. When deployed in practice, this premise fails under three critical conditions:

- **Commit-and-Forget Allocation:** Once a model is selected, the system is locked into that choice for the duration of the response. There is no mechanism to detect when a smaller model hits a reasoning bottleneck mid-generation, and transitioning to a more capable model requires restarting generation entirely. A single misrouted prompt can therefore produce a confidently wrong output rather than triggering a graceful escalation.

- **Prompt-Level Blindness:** Routing decisions are executed over the entire prompt context prior to the generation of a single token, forcing a uniform computational strategy across a highly uneven task. A prompt requiring simple context retrieval followed by complex, multi-step deduction is constrained to a single model assignment that suboptimally serves both operations. The fine-grained, token-level adaptivity achieved by micro-routing architectures remains fundamentally inaccessible at this granularity.

- **Router Self-Interference:** The routing mechanism itself introduces computational latency. A heavy classifier or uncertainty estimator can easily negate the savings it was designed to capture, particularly for simple queries directed to smaller models. In these cases, the overhead of the routing decision paradoxically increases the total operational cost.

Ultimately, prompt-level routing remains structurally incapable of the dynamic coordination required for complex, multi-stage tasks. To address these static limitations, Section 4.3 examines collaborative architectures that move beyond simple selection toward multi-agent frameworks for continuous, distributed reasoning.

### 4.3 Multi-Agent Collaborative Hybrids (Structural-Level)

Whereas the previous categories focus on internal decoding loops or prompt-level routing, this category addresses how inference is structurally organized across multiple models operating simultaneously. Foundational collaborative approaches operate on the premise that an ensemble of diverse models can error-correct itself through layered architecture or peer feedback. Mixture-of-Agents (MoA) (Wang et al., 2024) establishes this collaborative paradigm by feeding models the outputs of other models in a layered structure. Similarly, multi-agent debate frameworks (Du et al., 2024) and Exchange-of-Thought (Yin et al., 2023) demonstrate

Table 4: Macro-Level Routing Methods

| Method | Routing Mechanism | Model Pool | Baseline Type | Cost Reduction | Quality Metric |
|---|---|---|---|---|---|
| FrugalGPT (Chen et al., 2023b) | Learned LLM cascade | 14 LLM APIs across 6 providers | Strong-model-only | Up to 98% cost reduction vs. GPT-4 alone | Matches or exceeds GPT-4 accuracy; +4% accuracy at same cost on HellaSwag, HEAD-LINES, OVERRULING, CoQA |
| RouteLLM (Ong et al., 2025) | Trained preference classifier | GPT-4 / Mixtral-8x7B | Strong-model-only | Over 85% on MT-Bench; 45% on MMLU; 35% on GSM8K | 95% of GPT-4 performance retained; APGR >50% over random baseline |
| AutoMix (Aggarwal et al., 2024) | Few-shot self-verification + POMDP meta-verifier | Mistral-7B / LLaMA-2-13B / GPT-3.5 | Strong-model-only | Over 50% cost reduction for comparable performance | Consistently positive incremental benefit per cost ($\Delta$IBC) across CNLI, QASPER, NarrativeQA, CoQA |
| Uncertainty Routing (Zhang et al., 2025) | Semantic entropy confidence signal | GPT-4 / Mixtral-8x7B | Strong-model-only | Lowest API cost at \$3.74 vs. \$4.06 random at CPT(80%) on MT-Bench | Superior response quality vs. RouteLLM and TO-Router on MT-Bench and GSM8K |
| CP-Router (Su et al., 2025a) | Conformal Prediction uncertainty | LLaMA-3.1-8B / DSR1-LLaMA-8B; Qwen-2.5-14B / DSR1-Qwen-14B | Strong-model-only | Substantial token reduction vs. routing all to LRM | Highest token utility on all benchmarks with LLaMA pairing; 5/6 with Qwen pairing on GPQA, MMLU, CN-Chemistry |

*Notes:* The Baseline Type column indicates the primary comparison standard used in each paper. "Strong-model-only" denotes a baseline in which all queries are routed to the most capable (and expensive) model in the pool. Hybrid LLM (Ding et al., 2024), FORC (Šakota et al., 2024), Expert Router (Pichlmeier et al., 2024), Routing to the Expert (Lu et al., 2024), and LLM Routing with Benchmark Datasets (Shnitzer et al., 2023) are omitted because they do not report standardized cost reduction figures against a consistent strong-model-only baseline. Cascade Routing (Dekoninck et al., 2025), MetaLLM (Nguyen et al., 2024), and MixLLM (Wang et al., 2025b) are excluded because their reported gains reflect theoretical optimality conditions or relative improvements over other routing methods rather than absolute cost reduction against a strong-model-only baseline.

that allowing models to propose, argue, and exchange partial reasoning traces significantly outperforms single-model reasoning. The logic driving this trajectory is empirically supported by scaling laws showing that sampling more agents and applying majority voting consistently improves performance on complex tasks (Li et al., 2024a). However, these methods suffer from a severe computational penalty: the cost of verification typically scales linearly, $O(A)$ (where A is the number of agents), with the number of agents.

The most direct response to this cost problem is to integrate speculative acceleration into the collaborative process itself, ensuring that heavy multi-agent compute is reserved strictly for verification rather than generation. A primary example of this synthesis is Speculate-then-Collaborate (CoSD) (Wang et al., 2025e), which employs a rule-based or tree-based decision function that governs when to invoke an assistant model to improve the draft, fusing complementary domain knowledge efficiently without retraining. Directly addressing the linear verification cost, Collaborative Decoding via Speculation (CoS) (Fu et al., 2025a) applies speculative decoding to multi-model collaborative decoding. In this framework, each model alternates between proposer and verifier roles, with theoretical guarantees that the speculative ensemble is never slower

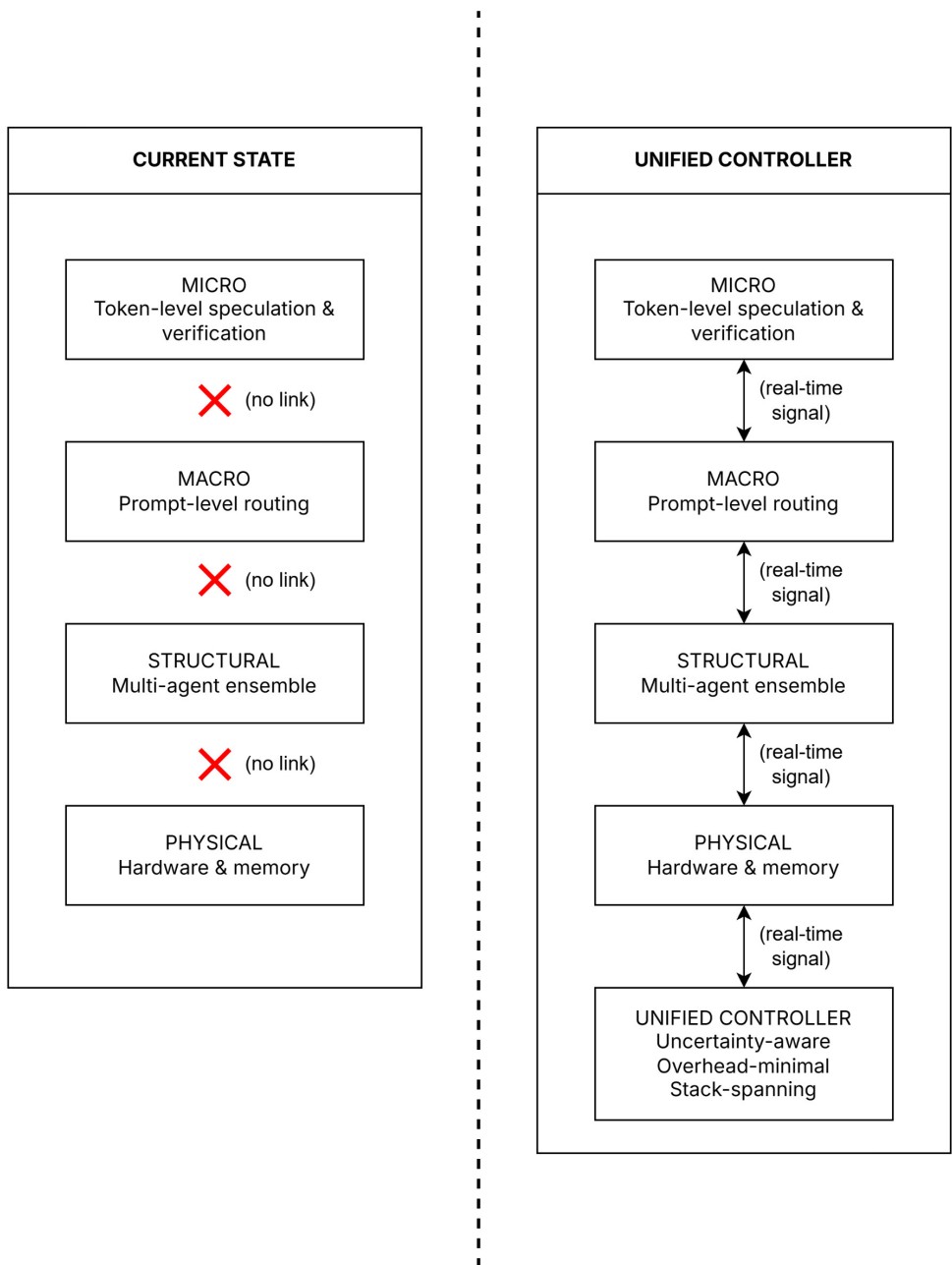

Figure 5: The current state (left) shows four isolated hybrid levels with no inter-level communication, whereas the proposed unified controller (right) connects all levels through bidirectional real-time signals, enabling a single uncertainty-aware, overhead-minimal, stack-spanning control layer.

than standard collaborative decoding. Mixture of Attentions for Speculative Decoding (Zimmer et al., 2025) takes this further, introducing a small-model architecture using cross-attention to the target model's hidden states for drafting, enabling a client-server deployment where the draft model runs locally on a consumer device while verification is offloaded to a remote server.

Rather than changing how drafts are generated, a second direction attacks the same cost problem by controlling which agents participate and when. HISPEC (Kumar et al., 2025) organizes the pipeline into a strict three-tier hierarchy of draft model, intermediate verifier, and target model, ensuring inaccurate draft tokens are rejected at the intermediate layer before invoking full target verification. Complementing this, Automatic Task Detection (Ge et al., 2025) extends heterogeneous speculative decoding by automatically classifying the incoming task type to dynamically assign the most suitable draft model, avoiding the overhead of a one-size-fits-all drafter. At the multi-agent level, frameworks like MacNet (Qian et al., 2025) organize agents into a directed acyclic graph (DAG) topology to optimize collaborative scaling. To directly reduce the computational overhead of these ensembles, MARS (Wang et al., 2025c) eliminates costly reviewer-to-reviewer interactions by having reviewer agents evaluate independently in parallel, with a meta-reviewer aggregating their confidence scores into a final decision, reducing token usage and inference time by approximately 50% compared to standard multi-agent debate. Extending this dynamic participation, Multi-Agent Collaboration via Evolving Orchestration (Dang et al., 2025) addresses the rigidity of fixed agent participation by training a centralized orchestrator via reinforcement learning to dynamically activate and deactivate agents based on the current state of the task.

At the finest granularity, the participation question reduces to the token level, where the system must decide not which agents to activate but whether to invoke another model at all. Learning to Decode Collaboratively (Shen et al., 2024) addresses this through bilateral token-level collaboration, training a single routing layer on top of the base model's hidden states to decide at each step whether to continue locally or invoke an assistant, significantly reducing unnecessary model invocations. As these architectures grow in complexity, researchers have formalized the scaling behavior of compound multi-model inference (Chen et al., 2024a), analyzing exactly when additional LLM calls yield diminishing returns and establishing theoretical limits on sub-linear efficiency gains.

**Synthesis.**  These structural-level hybrids represent a meaningful step toward making multi-agent inference practically deployable. Delegating generation to lightweight drafters while reserving heavier models for verification and consensus reduces the computational burden that makes pure ensemble approaches prohibitive. Despite the efficiency gains, the underlying participation logic remains insufficiently responsive to task dynamics, and several structural inefficiencies persist:

- **Uniform Agent Taxation:** Most collaborative frameworks invoke the full ensemble regardless of whether a given generation step warrants it. Simple tokens within a complex prompt incur the same verification cost as genuinely ambiguous ones. Until agent participation is gated on real-time difficulty rather than fixed topology, the efficiency floor of these systems remains tied to their most expensive configuration.

- **Persistent Verification Scaling:** Even when speculative drafting reduces generation costs, the cost of verification continues to scale with the number of participating agents. For ensembles beyond a small handful of models, this linear growth ($O(A)$) creates a hard efficiency barrier that clever drafting strategies alone cannot overcome. MARS's (Wang et al., 2025c) independent parallel evaluation is a partial solution, but one that trades consensus richness for throughput.

- **Consensus Latency:** Frameworks requiring peer agreement before committing a token or step are bottlenecked by their slowest participant, and this latency compounds across long generation sequences. Asynchronous participation and partial-consensus mechanisms remain underexplored relative to the theoretical gains they could unlock.

The consensus latency and synchronization overhead that persist even in the most optimized collaborative topologies ultimately reveal that the bottleneck is no longer purely algorithmic. Resolving it requires the hardware-aware rethinking that Section 4.4 takes as its starting point.

### 4.4   Distributed & Hardware-Aware Hybrids (Physical-Level)

The final level of hybridization occurs at the physical and system level. Standard speculative and reasoning algorithms typically assume a uniform cost model for token generation, but in real deployments, latency

varies substantially depending on model architecture, hardware state, and memory bandwidth. Hardware-aware hybrids address this by integrating system-level constraints directly into the speculation and reasoning policy, through two distinct mechanisms: altering the speculative policy based on real-time hardware signals, and restructuring the physical topology of the inference infrastructure across different devices or compute phases.

The first mechanism, policy-level adaptation, is most prominently driven by the specific requirements of Mixture-of-Experts (MoE) architectures, where standard speculative decoding introduces a unique memory bottleneck. When speculating a sequence of tokens, each token may activate a different expert subset; for example, Token 1 may activate Experts A and B, while Token 2 activates Experts C and D. This forces the system to continuously load and unload parameters from memory, saturating bandwidth and potentially negating speculative speedups. Cascade (Chen et al., 2024c) addresses this by implementing a utility-based cost model that calculates the ratio of time saved to expert-loading latency, using a hill-climbing heuristic to dynamically set the speculation length. SpecMoEOff (Wang et al., 2025g) and PowerInfer (Song et al., 2024) take a different approach, managing VRAM limitations through heterogeneous offloading, identifying frequently accessed hot parameters for high-bandwidth GPU memory while pushing cold parameters to system RAM via asynchronous pipelining.

Policy-level adaptation also extends to device-level scheduling and topology, where drafting and verification are distributed across heterogeneous or remote systems. Mirror Speculative Decoding (Bhendawade et al., 2025) and DuoDecoding (Lv et al., 2025) decouple the draft–verify loop across separate devices, such as GPU–NPU combinations, reframing speculation as a scheduling problem in which throughput gains arise from maximizing parallel device utilization. In cross-network environments, Distributed Speculative Decoding (DSD) (Yu et al., 2025), Distributed Split Speculative Decoding (DSSD) (Ning et al., 2025), and Speculative LLM Edge Decoding (SLED) (Li et al., 2025b) distribute the draft and target models across physical locations, placing a lightweight drafter on the edge while delegating verification to the cloud, addressing the memory limitations of resource-constrained hardware. Fast Inference via Hierarchical Speculative Decoding (Mohri et al., 2025) and Sequoia (Chen et al., 2024b) formalize the interaction between candidate token tree structures and hardware budgets, constructing trees whose branching depth and verification strategies are explicitly chosen to align with available memory bandwidth. However, executing these tree-structured verification passes efficiently requires more than policy-level tuning. Standard attention kernels are designed for dense, sequential memory access patterns, and tree-structured verification degrades arithmetic intensity by forcing irregular, low-reuse memory access across divergent candidate branches, falling far below the ridge point (Williams et al., 2009) where GPU compute units can be fully utilized. Resolving this requires operator fusion—combining each operation's substeps into dedicated custom CUDA kernels to eliminate intermediate HBM round-trips within attention, RMSNorm, and MLP, respectively—thereby recovering the memory bandwidth efficiency that tree-structured topologies inherently disrupt.

The second mechanism, infrastructure topology restructuring, takes a different starting point entirely, changing where inference computation happens rather than how the speculative policy behaves. DistServe (Zhong et al., 2024) and Splitwise (Patel et al., 2024) introduced the principle of disaggregating the prefill and decoding phases onto separate specialized GPUs. Because the prefill phase is compute-bound whereas the decoding phase is memory-bound, treating them as a monolithic pipeline underutilizes both hardware types. SARATHI (Agrawal et al., 2023) refines this further by piggybacking decode requests onto prefill batches, filling the idle compute cycles that arise during memory-bound decoding with useful prefill computation, prioritizing hardware-specific utilization over traditional linear inference execution.

**Synthesis.** Physical-level hybrids treat hardware topology as a design variable rather than a fixed constraint. Where prior categories operate within a given hardware environment, this category makes the hardware itself configurable, disaggregating computation across devices, memory tiers, and network boundaries. However, this shift introduces its own set of limitations:

- **Semantic Opacity:** The optimizations in this category treat the model as a computational black box. Disaggregating prefill and decode phases, offloading parameters, or distributing draft and verify across devices are indifferent to generation content. Unlike policy-level hybrids, these systems have no mechanism to recognize a pivotal reasoning step and allocate additional compute, or to

detect that a sequence has entered a low-entropy regime where speculation could be safely extended. Physical efficiency and semantic awareness remain entirely decoupled.

- **Synchronization as the New Bottleneck:** Decoupling drafting and verification across heterogeneous devices or edge-cloud splits introduces a new latency source: the cost of synchronizing KV caches across PCIe buses or network links. In constrained or high-latency environments, this communication overhead can exceed the compute savings that distribution was intended to provide, particularly when accepted token sequences require cascading cache updates across multiple physical locations.

- **Configuration Brittleness:** Frameworks built around specific memory budgets or device pairings degrade when deployed on hardware with different profiles. A system that co-designs speculative tree depth with available bandwidth is locally optimal but does not transfer across the heterogeneous hardware environments that characterize real production infrastructure. No single physical-level configuration generalizes.

The fact that token-level, prompt-level, structural, and physical hybridization have each failed to independently close the speed–reasoning tradeoff points to a fundamental gap that no single level can address in isolation. The field's next step must be a unified controller operating across all four levels simultaneously, which is the core challenge formalized in Section 5.1 as the Orchestration Gap.

## 4.5 Cross-Category Methods

While the four-level taxonomy presented above provides a necessary structural framework, several architectures surveyed in Sections 4.1–4.4 resist clean placement within a single level. These systems are significant not as edge cases but as indicators: each one exploits a different seam in the taxonomy, revealing where the boundaries of the framework are structurally fluid.

The most heavily populated boundary is the micro–macro seam, which features systems that execute coarse-grained model allocation at fine-grained resolutions. For instance, CITER (Zheng et al., 2025) and Speculative Thinking (Yang et al., 2025b) operate firmly within the token- or segment-level generation loop, characteristic of the micro level. Yet, they execute the kind of model-switching and escalation decisions traditionally reserved for prompt-level macro routers. RelayGen (Song et al., 2026a) deliberately compromises between the two, operating at segment granularity to apply macro-level routing logic without disrupting the contiguous draft spans required by micro-level speculation. Similarly, EASD (Su et al., 2025b) bridges this gap by sitting inside the speculative generation loop while relying on global distributional entropy—an uncertainty signal characteristic of macro-level routing—to trigger resampling.

The macro–structural seam surfaces in frameworks that combine prompt-level routing with limited multi-model coordination or self-supervised ensemble dynamics. AutoMix (Aggarwal et al., 2024) functions primarily as a macro-level router, but its few-shot self-verification creates a two-stage ensemble dynamic that approximates structural-level collaboration without incurring the full computational cost. CP-Router (Su et al., 2025a) routes between models at the macro level but relies on conformal prediction uncertainty computed over the model's own outputs. This approach blurs the line between self-supervised evaluation and structural multi-model selection.

Finally, the micro–structural seam is defined by systems where token-resolution generation loops directly dictate multi-agent participation. Learning to Decode Collaboratively (Shen et al., 2024) spans this boundary by operating at token resolution to dynamically govern which assistant models participate in the ensemble. HISPEC (Kumar et al., 2025) approaches this intersection from the opposite direction: it employs a structural three-tier agent hierarchy, but its intermediate verifier operates via early exits from the target model itself, functionally mirroring self-speculative micro-level mechanisms.

Taken together, these overlaps reveal a consistent pattern: the boundaries of the taxonomy correspond to unresolved design tradeoffs rather than clean architectural distinctions. The micro–macro seam is contested by the granularity of the routing decision; the macro–structural seam is contested by the degree of inter-model coordination; and the micro–structural seam is contested by whether multi-model participation is

governed token-by-token or at a coarser level. None of the four levels provides the mechanisms needed to resolve all three simultaneously, which forms the structural argument for the unified controller formalized in Section 5.1.

## 5    Research Gaps and Future Directions

The taxonomy presented in Section 4 demonstrates that the field of LLM deployment is rapidly moving beyond isolated latency and quality optimizations. The emergence of micro-, macro-, structural-, and physical-level hybrids represents significant progress, yet bridging the Orchestration Gap remains an open challenge. Beyond this, we identify four additional structural and evaluative challenges that must be addressed to fully realize orchestrated inference.

### 5.1    The Orchestration Gap (Unified Control)

**The Challenge.**  No unified control mechanism currently exists that can dynamically regulate compute allocation across the full inference stack. The four hybrid levels surveyed in Section 4 each partially address this: micro-level systems switch between models at the token level, macro-level routers allocate compute at the prompt level, structural ensembles distribute verification across agents, and physical-level systems adapt to hardware constraints. However, each operates independently and within fixed boundaries, with no cross-level coordination. As a result, computational depth remains largely predetermined, leaving models prone to over-computing on simple queries and under-resourcing complex ones.

**Future Direction.**  Bridging this gap requires a dynamic control layer that unifies all four levels (micro, macro, structural, and physical), adjusting compute allocation in real time based on the difficulty of what is being generated. While individual training-free mechanisms such as EASD (Su et al., 2025b) at the micro-level and CP-Router (Su et al., 2025a) at the macro-level represent promising steps toward this goal, each addresses only a single level in isolation, with no mechanism to propagate signals across level boundaries. Future research must move beyond static configurations toward closed-loop controllers that treat inference as a dynamic decision process spanning the full stack, using signals such as entropy or gradient norms to trigger deeper reasoning only when warranted.

### 5.2    Operational Rigidity and the Need for Hierarchy

**The Challenge.**  Current structural speculation methods and multi-agent frameworks often operate on flat computational topologies. They follow fixed branching schemes or static agent participation rules that remain unchanged throughout generation, leading to linear verification costs ($O(N)$). These fixed structures cannot adapt to the varying difficulty of natural language. During high-entropy phases, static trees waste compute on unnecessary branches; during low-entropy phases, parallel capacity sits underutilized.

**Future Direction.**  Research should move toward hierarchical inference structures. While architectures like HISPEC (Kumar et al., 2025) successfully introduce a three-tier hierarchy by exploiting early-exit layers within a single target model for draft generation and intermediate verification to reject inaccurate tokens early, the limitation is that these exit points are fixed before inference begins. The next natural step is making these hierarchies fully dynamic. Rather than fixing tree shape or agent participation in advance, the system should dynamically route computation to expand or prune branches based on real-time step difficulty. We propose lightweight global managers capable of decomposing tasks and assigning them to specialized mid-level agents, which in turn dynamically control their own local speculative drafters.

### 5.3    KV Cache Efficiency in Branching

**The Challenge.**  The transition from standard autoregressive decoding to non-linear generation via speculative trees or multi-agent ensembles fundamentally disrupts memory access patterns. In standard decoding, the KV cache grows linearly and remains contiguous. Branching methods, by contrast, require the system

to maintain multiple divergent histories simultaneously, leading to non-contiguous memory access and significant duplication of KV states (Miao et al., 2024; Yao et al., 2025). While PagedAttention (Kwon et al., 2023) handles dynamic branching via copy-on-write block sharing, the real limitation lies at the kernel level: attention kernel partitioning is suboptimal for tree-structured access patterns, causing redundant KV cache IO between GPU global memory and shared memory, poor load balancing across verification passes, and a lack of memory access reuse for shared prefixes across divergent branches (Yao et al., 2025).

**Future Direction.**  Future research must prioritize speculation-aware memory management, with new paging algorithms and cache-sharing strategies designed specifically for branching access patterns. Rather than allocating cache uniformly across all speculative branches, these systems should weight memory allocation based on generation probability, proactively loading high-probability branch data into on-chip cache before it is needed. Alongside this, zero-copy branching architectures would allow multiple speculative paths to reference a unified shared context without physically duplicating data across PCIe buses or network links.

## 5.4  Standardization of Efficiency-Reasoning Metrics

**The Challenge.**  The community currently lacks a unified metric for evaluating the tradeoff between latency and reasoning quality. Inference acceleration literature typically prioritizes throughput and speedup, treating output quality as a pass/fail constraint rather than a graded variable (Xia et al., 2024b; Yan et al., 2024; Hu et al., 2025a). Reasoning-focused research, conversely, emphasizes benchmark accuracy or Pass@$k$ while neglecting the computational cost required to achieve those scores (Wang et al., 2023; Chen et al., 2021; Li et al., 2024a). This disconnect makes it impossible to objectively compare a fast, shallow model against a slower, deeper reasoner.

**Future Direction.**  Future benchmarks should adopt compound evaluation metrics that measure both dimensions simultaneously. We propose two unified metrics: Correctness Per Second (CPS), which measures the rate at which a system generates factually and semantically verified tokens; and Inference-Time-to-Solution (ITTS), which measures the end-to-end wall-clock time required to reach a verified consensus on a complex task. Both metrics penalize reasoning strategies that yield marginal quality gains at exponential latency costs, pushing evaluation toward the most efficient path to a correct answer. Formal mathematical definitions of CPS and ITTS are provided in Section 6.1.5.

## 5.5  The Auxiliary Model Overhead

**The Challenge.**  A recurring structural limitation across the micro-level and structural-level hybrids discussed in Sections 4.1 and 4.3 is a heavy reliance on auxiliary components. Integrating Process Reward Models (PRMs) (Lightman et al., 2023), reinforcement-learning-based token routers like CITER (Zheng et al., 2025), and separate evaluator modules like Judge Decoding (Bachmann et al., 2024) fundamentally increases deployment friction, VRAM overhead, and system fragility. Furthermore, these auxiliary models must be perfectly aligned with the target task and continuously updated, introducing a prohibitive training complexity that prevents seamless deployment.

**Future Direction.**  Future research should prioritize mechanisms that govern speculation, routing, and verification using signals already available during generation. Training-free and self-supervised approaches offer the lowest deployment friction, with frameworks like SelfJudge (Yoon et al., 2025), which derives supervision from the target model's own bidirectional generation behavior, and EASD (Su et al., 2025b), which triggers resampling from entropy signals alone, representing promising steps in this direction. That said, trained auxiliary components remain justified when deployment volume amortizes their cost, as methods like Judge Decoding (Bachmann et al., 2024) and CITER (Zheng et al., 2025) can demonstrate substantially superior performance when training-domain alignment is maintained. The broader goal is not to eliminate auxiliary models categorically, but to ensure their complexity is proportionate to deployment constraints.

# 6 Evaluation Metrics and Datasets

Static evaluation metrics such as raw speedup for speculative decoding or raw accuracy for reasoning frameworks are insufficient for hybrid inference, where performance depends not just on speed or quality in isolation but on how well a system manages the tradeoff between them. Because orchestrated systems dynamically balance fast and deep computation across varying hardware topologies, their evaluation must account for latency spikes, routing overhead, memory contention, and reasoning stability. This section outlines the metrics and datasets needed to evaluate these systems comprehensively.

## 6.1 Evaluation Metrics

### 6.1.1 Speculative and Micro-Level Efficiency

While raw token-per-second throughput is a standard measure of speed, it obscures the internal mechanics of draft-target alignment in micro-level hybrids.

- **Speedup Ratio ($S$):** The foundational metric for micro-level hybrids (Leviathan et al., 2023), measuring the wall-clock latency of standard autoregressive decoding divided by the latency of the speculative hybrid.

- **Acceptance Rate ($\alpha$) and Average Acceptance Length ($\tau$):** $\alpha$ is a foundational field-standard metric measuring the probability that a target model accepts a drafted token, but degrades sharply on high-entropy tasks. $\tau$ complements this by averaging the number of draft tokens accepted per verification round, capturing how acceptance holds up across longer reasoning sequences (Li et al., 2024d).

- **Draft Length ($\gamma$) and Optimal Draft Budget:** $\gamma$ controls the number of tokens the draft model proposes per step, and its optimal value is task- and hardware-dependent (Lv et al., 2025). For tree-based and hierarchical methods (see Sections 3.1 and 4.4), the optimal draft budget must be tuned jointly against acceptance rate, because over-drafting causes tokens positioned later in the draft sequence to exhibit diminished acceptance rates, wasting verification compute without improving throughput (Lv et al., 2025).

- **Token Generation Throughput and Wall-Clock Latency:** Beyond single-sequence speedup, serving-oriented evaluation requires measuring tokens generated per second across concurrent requests (Kwon et al., 2023). Wall-clock latency captures end-to-end user-facing delay, including batching and memory overhead, which speedup ratio alone does not reflect.

### 6.1.2 Serving, Infrastructure, and Physical Dynamics

The transition from linear autoregressive decoding to hardware-aware, disaggregated generation fundamentally changes physical execution, making infrastructural constraints just as important as algorithmic efficiency. The metrics in this subsection are particularly critical for the disaggregated and heterogeneous systems discussed in Section 4.4.

- **Time to First Token (TTFT) and Time Per Output Token (TPOT):** TTFT isolates the compute-bound prefill phase, while TPOT captures the memory-bound decode phase (Zhong et al., 2024). These are critical counter-metrics for physical-level disaggregation, where raw throughput masks user-facing latency.

- **Goodput and SLO Attainment Rate:** Goodput quantifies the usable capacity of a system under strict service-level-objectives (SLOs) for both TTFT and TPOT, penalizing instability (Zhong et al., 2024). Alongside it, the SLO Attainment Rate measures the exact fraction of requests that successfully meet both constraints simultaneously, capturing the true trade-off between algorithmic efficiency and system-level queuing delays (Zhong et al., 2024).

- **Time Between Tokens (TBT):** While TPOT averages decode latency across a full output sequence, TBT measures the per-iteration latency between consecutively generated tokens under a fixed token budget (Agrawal et al., 2023). It is a critical diagnostic metric for hybrid batching systems, where dynamic scheduling decisions can cause inter-token stalls that TPOT's averaging effect does not expose.

- **KV Cache Memory Utilization:** This metric captures the GPU memory consumed by KV cache states during serving, exposing how fragmentation and redundant duplication of KV entries constrain batch size and limit overall system throughput (Kwon et al., 2023). It is critical for disaggregated deployments, where inefficient KV cache management directly reduces the number of requests that can be served concurrently.

### 6.1.3   Routing Economics and Collaboration Overhead

When inference is distributed across multiple models or agents, evaluation must measure the friction and financial cost of the orchestration layer itself.

- **API Cost and Router Overhead:** In macro-level routing systems (see Section 4.2), API cost per query directly measures the economic benefit of routing to cheaper models (Chen et al., 2023b; Ong et al., 2025). Router Overhead is a standard wall-clock measurement of the latency introduced by the routing decision itself, captured as the additional time-to-first-token incurred before generation begins. Both must be evaluated together to ensure the orchestration layer does not negate its own savings.

- **CPT and APGR:** Call-Performance Threshold (CPT) measures the fraction of strong-model calls required to achieve a specific quality baseline, such as 95% of target performance (Ong et al., 2025). Whereas CPT captures the cost side of this tradeoff, Average Performance Gain Recovered (APGR) measures how much of the performance gap between the weak and strong models the router closes (Ong et al., 2025). Together, they map the efficiency of the router's cost-performance tradeoff.

- **Token Cost per Response and Inference Time Reduction (%):** In structural multi-agent frameworks (see Section 4.3), Token Cost serves as the primary efficiency counter-metric, capturing the aggregate computation consumed by all collaborative agents to produce a final consensus. To capture the relative efficiency of novel multi-agent topologies, Inference Time Reduction evaluates latency or token usage savings against standard multi-agent debate baselines (e.g., the ~50% reduction demonstrated by MARS (Wang et al., 2025c)).

### 6.1.4   Reasoning Stability and Quality Constraints

When fast heuristic drafters or dynamic routers (see Section 4.2) are used to accelerate complex problem solving, evaluation must ensure that acceleration does not corrupt the underlying reasoning logic.

- **Task Accuracy and Pass@$k$:** Used heavily in mathematical reasoning and code generation (Chen et al., 2021; Cobbe et al., 2021), these metrics ensure that semantic and reward-guided verification mechanisms (see Section 4.1) preserve logical integrity. In hybrid inference, they act as strict constraints rather than variables to maximize.

- **Pass@$k$ and Cons@$k$:** Pass@$k$ measures the probability that at least one of $k$ sampled outputs is correct, serving as the standard correctness baseline for reasoning tasks (Chen et al., 2021). Cons@$k$ extends this by evaluating whether reasoning paths converge reliably across those samples. In hybrid systems, high Pass@$k$ combined with a low Cons@$k$ indicates brittle reasoning, revealing that the fast drafting mechanism or dynamic router has destabilized the model's logical consistency.

- **LC Win Rate and LLM-as-Judge:** For open-ended generation and multi-agent debate (see Section 4.3), exact-match metrics fail. Length-Controlled (LC) Win Rates (Dubois et al., 2024) and automated judges (Zheng et al., 2023) leverage frontier models to evaluate factuality and coherence, ensuring that fast drafting mechanisms do not degrade conversational quality.

### 6.1.5 Compound Trade-off Metrics

To bridge the Orchestration Gap (see Section 5.1), metrics that explicitly evaluate the intersection of latency and reasoning capability must be adopted, penalizing systems that incur exponential costs for marginal quality gains.

**Correctness Per Second (CPS).** As proposed in Section 5.4, this compound metric quantifies the efficiency of an inference path by measuring the rate at which a system produces verified correct outputs per unit of wall-clock time:

$$\text{CPS} = \frac{\sum_{i=1}^{N} \mathbf{1}[\hat{y}_i = y_i]}{T_{\text{wall}}} \tag{3}$$

where $\mathbf{1}[\hat{y}_i = y_i]$ is a binary correctness indicator for each evaluated output, $N$ is the total number of evaluated responses, and $T_{\text{wall}}$ is the total wall-clock inference time in seconds. This formulation objectively compares a fast, shallow hybrid against a slow, deep reasoner without privileging either dimension in isolation.

**Inference-Time-to-Solution (ITTS).** This metric measures the end-to-end wall-clock time required to produce the first verified correct output on a complex task:

$$\text{ITTS} = T_{\text{prefill}} + T_{\text{decode}} + T_{\text{sync}} \tag{4}$$

where $T_{\text{prefill}}$ is the compute-bound prompt processing time, $T_{\text{decode}}$ is the memory-bound token generation time, and $T_{\text{sync}}$ captures synchronization overhead introduced by routing decisions or multi-agent consensus mechanisms. The clock stops at the first verified correct output, naturally penalizing over-deliberation and incentivizing systems to optimize for the most efficient path to a correct answer rather than maximizing raw speed or accuracy in isolation.

### 6.2 Evaluation Datasets

Evaluating hybrid inference systems requires a diverse portfolio of datasets that test both the algorithmic stability of the model and the physical resilience of the serving infrastructure. While static LLMs are typically evaluated on pure reasoning capability, orchestrated hybrids must also be stress-tested against real-world prompt distributions, fluctuating sequence lengths, and high-entropy edge cases that trigger dynamic routing or multi-agent escalation. Table 5 outlines the primary datasets used across the literature to validate these architectures.

## 7 Conclusion

As large language models shift from research prototypes to production environments, the conflict between inference latency and reasoning capability has emerged as a central challenge. This survey reviews the recent convergence of these two previously isolated domains, mapping out a taxonomy of Hybrid Inference architectures across four distinct levels: Micro (token-level speculation), Macro (model routing), Structural (multi-agent collaboration), and Physical (hardware-aware disaggregation).

Our analysis reveals that these hybrid techniques have individually advanced considerably, producing faster drafts and deeper reasoning capabilities. However,current systems remain constrained by localized optimizations, static routing policies, and rigid computational hierarchies that cannot adapt to prompt complexity at runtime. This lack of holistic flexibility results in a persistent Orchestration Gap, wherein systems cannot seamlessly scale their compute depth across the entire inference stack to match the complexity of the prompt.

We conclude that the next frontier of LLM inference lies not in optimizing these hybrid levels in isolation, but in truly universal inference. Future systems must combine the efficiency of speculative decoding with

Table 5: Evaluation Datasets for Hybrid Inference Systems

| Dataset / Corpus | Evaluation Domain | Primary Hybrid Level | Representative Papers |
|---|---|---|---|
| MT-Bench (Zheng et al., 2023), Arena-Hard (Li et al., 2024b), AlpacaEval 2.0 (Dubois et al., 2024) | Open-Ended Dialogue & Instruction Following | Micro (4.1), Macro (4.2), Structural (4.3) | MoA (Wang et al., 2024), Judge Decoding (Bachmann et al., 2024), RouteLLM (Ong et al., 2025) |
| GSM8K (Cobbe et al., 2021), MATH-500 (Hendrycks et al., 2021b), AIME 2025, GPQA Diamond (Rein et al., 2024) | Symbolic & Multi-Step Deductive Reasoning | Micro (4.1), Macro (4.2), Structural (4.3) | SpecReason (Pan et al., 2025), SpecCoT (Wang et al., 2025a), SelfJudge (Yoon et al., 2025), RouteLLM (Ong et al., 2025), Judge Decoding (Bachmann et al., 2024) |
| HumanEval (Chen et al., 2021), LiveCodeBench (Jain et al., 2025), MBPP (Austin et al., 2021) | Code Generation & Execution Correctness | Micro (4.1), Macro (4.2), Physical (4.4) | RouterBench (Hu et al., 2024), Judge Decoding (Bachmann et al., 2024), DSD (Lv et al., 2025), SelfJudge (Yoon et al., 2025) |
| MMLU (Hendrycks et al., 2021a), ARC Challenge (Clark et al., 2018), HellaSwag (Zellers et al., 2019) | Knowledge Retrieval & Out-of-Distribution | Micro (4.1), Macro (4.2) | RouterBench (Hu et al., 2024), RouteLLM (Ong et al., 2025), Judge Decoding (Bachmann et al., 2024) |
| CNN/DailyMail (Hermann et al., 2015), XSum (Narayan et al., 2018) | Summarization & Input-Constrained Generation | Micro (4.1), Physical (4.4) | DSDE (Yang et al., 2025a), SelfJudge (Yoon et al., 2025), DSD (Yu et al., 2025) |
| Azure LLM Traces (Patel et al., 2024), ShareGPT (Chiang et al., 2023) | Real-World Serving Workloads & Traffic Bursts | Physical (4.4) | DistServe (Zhong et al., 2024), SARATHI (Agrawal et al., 2023), Splitwise (Patel et al., 2024) |
| LongBench (Bai et al., 2024), WikiText-103 (Merity et al., 2017) | Long-Context Prefill & Tree Efficiency | Physical (4.4) | DistServe (Zhong et al., 2024), Sequoia (Chen et al., 2024b) |

*Notes:* Representative usage varies by paper; not all listed papers evaluate on all datasets within a given row. The papers listed illustrate the general evaluation domain based on their architectural focus.

the robustness of inference-time search, dynamically modulating computation in response to real-time uncertainty signals without the overhead of heavy auxiliary models. By bridging the gap between fast drafting and deliberate reasoning through unified control, inference can evolve from static decoding strategies to truly adaptive, context-aware computation.

# 8 Statement of Broader Impact

This survey synthesizes and organizes a rapidly growing body of research on hybrid LLM inference, aiming to lower the barrier for researchers and practitioners building systems that are both fast and capable. By identifying the Orchestration Gap and proposing unified evaluation metrics, such as CPS and ITTS, we aim to shift community incentives away from optimizing speed or quality in isolation and toward holistic system design. Positively, more efficient inference directly reduces the energy and hardware costs of deploying large language models, broadening access to capable AI systems for resource-constrained organizations and reducing the environmental footprint of serving infrastructure. Our taxonomy may also accelerate progress by helping researchers identify underexplored combinations of techniques across the four hybrid levels defined

in this work. However, we acknowledge potential negative implications. Lowering inference costs could accelerate the deployment of LLMs in contexts where their outputs are unreliable or harmful, and efficiency gains may disproportionately benefit well-resourced organizations that can implement complex multi-model orchestration pipelines. Furthermore, framing inference as an optimization problem over speed and quality may inadvertently deprioritize other important dimensions, such as fairness, safety, and interpretability. We encourage future work on orchestrated inference to incorporate these broader considerations into system design rather than treating them as orthogonal concerns.

We also note a second-order risk that is specific to the agenda this survey advances. By centralizing speed and reasoning quality as the axes of progress, this work—and the research it may catalyze—risks making efficiency the dominant lens through which inference systems are evaluated and funded. This framing is consequential: the compound metrics we propose in Section 5.4 capture correctness and latency, but omit fairness, robustness, and auditability—properties that become harder to assess as systems grow more complex and dynamically composed. These are not concerns orthogonal to orchestrated inference; they are properties that become harder to reason about as systems grow more complex and dynamically composed. We therefore encourage future work building on this taxonomy to treat safety, fairness, and interpretability as first-class design objectives rather than post-hoc constraints, and we call for the development of evaluation benchmarks that measure these properties in hybrid, multi-model settings with the same rigor currently applied to throughput and accuracy.

## 9 Limitations

As a survey paper, this work inherits several structural limitations that readers should consider when interpreting our taxonomy and analysis.

First, our four-level taxonomy (micro, macro, structural, physical) imposes discrete boundaries on what is, in practice, a continuous design space. Some methods span multiple levels simultaneously or resist clean categorization, and the boundaries we draw reflect organizational convenience as much as genuine architectural discontinuities. We explicitly discuss these overlapping frameworks and boundary-spanning architectures in Section 4.5.

Second, our coverage is necessarily shaped by publication availability. The field of hybrid inference is evolving rapidly, with many relevant works appearing as concurrent preprints during the writing of this survey. We have made a good-faith effort to include work available through early 2026, but we cannot guarantee completeness, and the landscape may have shifted materially by the time of publication. In particular, proprietary systems deployed by industry labs (e.g., the internal inference stacks behind commercial APIs) are largely invisible to our analysis, meaning that our taxonomy may underrepresent the state of practice relative to the state of published research.

Third, our proposed evaluation metrics (CPS and ITTS) remain theoretical constructs. We have not empirically validated them on a shared benchmark, and their practical utility depends on community adoption and the availability of standardized evaluation infrastructure that does not yet exist. The metrics also assume that correctness is binary and verifiable, which is a reasonable simplification for mathematical reasoning (Hendrycks et al., 2021b; Cobbe et al., 2021) and code generation (Chen et al., 2021; Jain et al., 2025) but a poor fit for open-ended generation tasks (Zheng et al., 2023; Dubois et al., 2024) where quality is graded and subjective.

Fourth, our analysis is predominantly architecture-centric and does not deeply engage with the training procedures, data requirements, or alignment considerations that shape the behavior of the component models within hybrid systems. A draft model trained on different data than its verifier, for example, may exhibit systematic biases that our taxonomy does not capture.

Finally, while we advocate for a unified controller spanning all four hybrid levels, we do not provide a concrete architectural proposal or formal specification for such a system. The Orchestration Gap, as we define it, is a diagnosis rather than a solution, and closing it will require contributions from systems, algorithms, and hardware co-design communities working in concert.

## 10 Reproducibility Statement

This is a survey paper and does not introduce new models, training procedures, or experimental results. Consequently, the primary reproducibility concern is the transparency and replicability of our literature search and selection methodology.

Our corpus was assembled through a systematic search of arXiv, Semantic Scholar, and the proceedings of major venues including NeurIPS, ICML, ICLR, ACL, EMNLP, and ACM SOSP/ASPLOS, using query terms spanning speculative decoding, inference-time computation, LLM routing, multi-agent inference, and hardware-aware serving. We included works published or publicly available as preprints up to the submission date. For preprints not yet peer-reviewed, we included only those deposited on arXiv with sufficient technical detail to assess their claims. Papers were included if they directly addressed the latency–quality tradeoff in LLM inference; works focused solely on training efficiency or model compression without an inference-time component were excluded.

The benchmark tables in Sections 3 and 4 report numbers directly as stated in the original papers. Where a paper reported results across multiple model sizes or configurations, we selected the figure most representative of the method's primary contribution, as described in that paper's abstract or main results section. Table footnotes document the specific exclusion criteria applied in each case to ensure the comparisons remain internally consistent.

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
