# OpenReview forum: "A Survey of Hybrid Inference Systems for Large Language Models"
_TMLR — Under review for TMLR_

### Review · Reviewer_ryqG · 2026-04-28

**Summary Of Contributions:**

This paper surveys hybrid inference systems for LLMs, which combines latency optimization such as speculative decoding, and reasoning optimization that utilizes methods like chain-of-thought to enhance the reasoning quality. These two concepts start as two isolated research trends, the paper suggests that these two topics are related and there is an orchestration gap. Recent studies have been sitting between these two trends and closing the gaps, i.e. combining the two approaches to improve the reasoning while optimizing the latency. The paper proposed a four-level taxonomy that classifies the recent studies. By surveying the recent research trends, the paper further discusses the future direction on orchestrated inference.

**Audience:**

Yes

**Audience Explanation:**

LLM inference, including the latency optimization and reasoning optimization, is a popular research topic nowadays.

**Broader Impact Concerns:**

No concern.

**Claims And Evidence:**

Yes

**Claims Explanation:**

The paper surveyed a large collection of recent studies. The paper argues the orchestration gap and suggests hybrid and orchestrated architecture should be studies in a taxonomy of four categories (sec 4), and surveyed and summarized the related works in each category and combined.

**Requested Changes:**

For better clarity, consider clarify and discuss the choice of the taxonomy of Hybrid and Orchestrated Inference architectures (Figure 4), specifically on how and why these four classes are chosen, and how there are related to each other.

---

> ### Author Response · Authors · 2026-06-06
> **Response to Reviewer ryqG**
>
> Thank you for this suggestion. We agree that the taxonomy in Figure 4 would benefit from a more explicit justification of how and why the four levels were chosen. The current text introduces the categories and defines them individually, but does not articulate the organizing principle that motivates the division. In the revision, we will add a short clarifying passage at the start of Section 4 explaining that the four levels are distinguished by the granularity at which compute allocation decisions are made: token-level for micro, prompt-level for macro, model-participation-level for structural, and infrastructure-level for physical. We will also clarify that the levels are not independent design choices but form a natural progression in the scope of orchestration, with each level addressing limitations that the previous one cannot resolve, a relationship that is implicit in the per-section synthesis discussions but deserves to be stated upfront. We believe this is a small addition that meaningfully improves the accessibility of the taxonomy.

---

### Review · Reviewer_6DA5 · 2026-05-14

**Summary Of Contributions:**

The paper reviews the interplay between speed improvements, performance improvements, and hardware optimizations in LLMs at inference time. The core contribution is that, while past reviews have studied these different aspects independently, they inherently cannot be disentangled: techniques for optimizing speed tradeoff performance, techniques for optimizing performance tradeoff speed, and the effectiveness of different hardware optimization techniques inherently depend on the techniques used to improve speed and performance.

Broadly speaking, the paper starts by considering techniques for speedups that are based on using small models to rollout proposals and evaluating the resulting subsequences in parallel using the larger model. For the performance angle, the paper considers methods for thinking/reasoning. In terms of hardware optimizations, the techniques reviewed primarily focus on distributed architectures and the cost of inter-device communication as well as loading to/from memory.

The paper then goes into methods which have attempted to tackle different scales of this problem simultaneously, which they call “hybrid” architectures. The main focus in this section is to discuss methods that try to *both* use smaller draft models when possible while also routing to reasoning mechanisms in large models or agents more optimally, such that the system as a whole stays closer to the pareto front of cost/performance tradeoffs. The scales considered are temporal in nature (a few tokens, entire sub-tasks) or architectural (e.g., how to interface networks of agents operating on potentially different hardware).

The authors then come to their views of how the field should progress. Broadly, they argue for more work that considers all levels of their discussed hierarchy: research into how various models can be networked together in order to make best use of different generation speeds, reasoning capabilities, and hardware parallelism. They don’t propose a specific architecture, but this is a review paper after all; the directions they propose are reasonable.

Finally, the authors discuss current evaluation metrics and datasets relevant to this field, and identify gaps in current evaluation metrics in light of their survey and the need for performance/speed tradeoffs.

**Audience:**

Yes

**Audience Explanation:**

I think so. I am not in this area of research, and I found the review pedagogical, wide-ranging, and accessible. I genuinely learned about new methods, techniques, and problems that I had not thought about beforehand. As an outsider, I may not be able to judge the interest of others who *are* in this subfield, but my sense is that it should be useful to them as well. So long as LLMs continue to be expensive to generate from, so long as they continue to be autoregressive, so long as they continue to be based on Transformers, and so long as reasoning remains the dominant paradigm for improving performance, I think that the methods and problems reviewed here will remain relevant. While I cannot evaluate the timescale over which the above paradigms will hold, in the near term at least I think this is paper could be useful to the field.

**Broader Impact Concerns:**

N/A, the authors have a broader impact statement and I think it is quite thoughtful and reasonable. On this note, the limitations section was also very well thought-out.

**Claims And Evidence:**

Yes

**Claims Explanation:**

This is a review paper, so answering questions about accuracy and clear evidence is a bit tricky. It’s worth saying at the outset that I am not familiar with this subfield, and have not thought about or come across many of the problems and methods that were discussed. That being said, I did find that these methods were clearly discussed and I found no evidence that the cited papers might be incorrectly represented. As far as I can tell, all the claims in the paper are correct and reasonable. The sections on future work in which the authors discuss what they see as the gaps in the field also seem thoughtful and reasonable to me — by reaching this point of the paper, I was beginning to have the same sorts of thoughts as well.

**Requested Changes:**

- My primary concern is that the paper is quite lengthy, and I think at times unnecessarily so. The same methods and metrics are repeatedly cited and discussed. Many of the discussed methods are discussed in what I think is a bit too much detail. In particular, I think that Section 4 is quite a bit longer than it needs to be, has the most redundancy, and discusses the most methods in overly fine-grained detail. I don’t see this length criticism as a nitpick: if a review paper is too long, it will end up being less frequently read. Even if those who do read it get a bit more out of it as a result of the extra detail, I think that the size of the audience will actually be more crucial to whether the review has a meaningful impact in the field. If the authors feel bad about removing detail perhaps they can consider adding an Appendix, but in any case I think the whole paper needs to be reread to address redundancy.
- Section 2.1 introduces what I found was a very nice and simple overall framing for the paper. I expected equation 1 to ground the discussion of many of the later discussed methods in a unified way. However, by the end of the paper it felt like this perspective of the underlying bottleneck was a bit isolated. If the discussion in Section 2.1 really is a unifying way to think about the different methods and the fundamental problem, perhaps the authors should consider referencing and deferring to it more often in their explanations. Otherwise, I think it can probably be removed.
- Several parts of the paper associate “difficulty” with “high-entropy” next-token predictions. While there may be an association here, I think the authors can be a bit more precise. In particular, high-entropy is not always associated with difficulty. For instance, the next token’s true distribution might simply have high-entropy because there is diffuse uncertainty (e.g., early tokens in a sequence). Indeed, a uniform distribution over all tokens would have the highest possible entropy, but I have no doubt that this distribution would be “easy” for the draft model to calibrate to. Really what “difficulty” would correspond to is the algorithmic complexity of the next-token distribution (i.e., something like its Kolmogorov complexity). I understand that treating this object would open up a whole can of worms that might distract from the paper’s main messages, but still, I think the authors should come up with some ways to specify what they mean when they say that a particular prediction is “difficult” for the draft model to make. Something like the KL divergence between the draft model and the large model might work (and the authors do discuss this), but of course this is tautological: we don’t know what the large model’s distribution is without evaluating it, which makes this metric of difficulty confusing. If there are no works that attempt to quantify the difficulty of the next token prediction using the draft model’s predicted distribution alone (other than entropy, which as I have said has clear problems), then perhaps the authors should more clearly emphasize the development of such a metric as an important area for future research. But certainly, I think they should highlight the problems of entropy that I have mentioned and add nuance to their discussions associating high-entropy predictions with “difficult” ones.
- I think that the discussion of evaluation metrics and datasets is a bit awkwardly placed. The authors just finished highlighting what is essentially their synthesized view of the current research and the important problems that the field should focus on, and subsequently go on to discuss the various metrics and datasets used in this area. I think a more natural place for this would be just prior to the section on research gaps and future directions.
- The section on evaluation datasets was a bit shallow, in my opinion. First, it is very short compared to the metrics section, but arguably just as important for being able to test these methods in practice. Second, I’m sure the authors, having reviewed all this work, can think of problems with the current datasets and benchmarks that are used in this area; identifying such problems could be quite important for people to develop novel datasets and benchmarks, which should be an important goal for a review paper.

---

> ### Author Response · Authors · 2026-06-06
> **Response to Reviewer 6DA5**
>
> Thank you for a thorough and constructive reading of the manuscript. We address each point in turn.
>
> &nbsp;
>
> 1. We agree that the length and redundancy concern, particularly in Section 4, is the most substantive issue raised, and we take it seriously. A survey that is too long to be read widely undermines its own purpose. We will conduct a targeted revision pass of Section 4 with the goal of consolidating overlapping method descriptions and reducing per-method detail where the marginal contribution to the reader's understanding is low. We are also receptive to the appendix suggestion as a way to preserve technical completeness without burdening the main narrative. We would, however, ask you to bear in mind that Section 4 covers the paper's core taxonomic contribution, and some degree of method-level detail is necessary to make the taxonomy actionable rather than purely abstract. We will aim to find the right balance rather than cutting uniformly.
>
> &nbsp;
>
> 2. Regarding Equation 1, our original intent was for the arithmetic intensity framing to serve as a conceptual anchor, but we acknowledge that in practice it is cited relatively sparingly after Section 2.1. We will revisit this in one of two ways: either by making the connection to Equation 1 more explicit when introducing methods in later sections (for example, when explaining why tree-structured speculation improves arithmetic intensity, or why disaggregated serving addresses the memory-bound regime), or by narrowing the claim about its unifying role in Section 2.1 to better reflect how it is actually used. We take your point that an under-used framing is worse than no framing.
>
> &nbsp;
>
>
> 3. On the use of entropy as a proxy for difficulty, you are correct that high entropy does not always correspond to difficulty in any meaningful sense. A diffuse but calibrated distribution over early sequence tokens is a valid example of high entropy that would be trivially easy for a draft model to handle. The deeper issue, as you note, is that the appropriate notion of difficulty is something closer to the divergence between the draft model's predicted distribution and the target's, which is unfortunately tautological without querying the target model. We will revise the relevant passages to distinguish between entropy as a local signal available to the draft model at inference time and difficulty as a property that fundamentally depends on draft-target alignment. We will also add explicit discussion flagging this as an open problem, noting that the field currently lacks a principled, target-model-free measure of next-token difficulty and that developing one would be a meaningful contribution. We believe this nuance strengthens rather than distracts from the paper's main messages.
>
> &nbsp;
>
> 4. The suggestion of moving Section 6 to immediately precede the research gaps discussion has structural merit, and we are open to making this change. Our original placement was motivated by wanting the evaluation discussion to feel grounded in the taxonomy that precedes it, but we recognize that the flow you describe, moving from taxonomy to gaps to evaluation tools to future directions, is arguably more natural. We will consider this reorganization in the revision.
>
> &nbsp;
>
> 5. Finally, the criticism regarding the depth of the datasets subsection is well-founded. The datasets discussion is notably shorter than the metrics discussion and does not perform sufficient analytical work. We will expand Section 6.2 to include observations about gaps in the current benchmark landscape: for instance, the near-absence of benchmarks that jointly stress-test latency and reasoning quality under realistic traffic distributions, the limited coverage of long-horizon multi-step tasks that would trigger dynamic routing or multi-agent escalation, and the lack of standardized evaluation protocols for hybrid systems across different hardware configurations. We believe this is one of the more actionable additions we can make to the revised manuscript.

---

> > ### Comment · Reviewer_6DA5 · 2026-06-07
> >
> > Thank you for engaging with the feedback. I look forward to seeing the changes addressing my and other reviewers' comments.

---

### Review · Reviewer_gsFh · 2026-05-27

**Summary Of Contributions:**

This paper surveys the emerging intersection of two historically separate lines of LLM inference research: latency optimization (primarily speculative decoding) and reasoning quality optimization (chain-of-thought, ensembles, search). The authors identify "Orchestration Gap," meaning the absence of a unified controller that can dynamically coordinate computation across all four levels simultaneously.

Strengths:
Section 4.5 honestly acknowledges where methods resist neat classification. The synthesis paragraphs at the end of each subsection are among the paper's strongest contributions, since they identify architectural ceilings of each level and give the reader a clear sense of why each level alone is insufficient. The comparison tables (Tables 2-4) are thorough and carefully annotated with exclusion criteria, which is good scholarly practice. The paper also covers a very large and recent body of work (many 2025-2026 preprints), making it timely.

Weaknesses:
Diagnosis but remains entirely aspirational that is no concrete architecture or even a formal specification is offered. Figure 3 is labeled as a conceptual illustration, not empirical data, which undermines claims about hybrid method behavior

**Audience:**

Yes

**Audience Explanation:**

The convergence of speculative decoding and inference-time reasoning is a live and practically relevant topic. Though the concepts are more aspirational than demonstrated correctly to be empirically proposed.

**Broader Impact Concerns:**

Section 8 is more substantive than most survey broader impact statements. It flags the risk that efficiency improvements could enable harmful deployments, acknowledges that CPS and ITTS ignore fairness and auditability, and raises the second-order concern that centering evaluation on speed and quality could crowd out safety-focused research.

**Claims And Evidence:**

No

**Claims Explanation:**

Paper's core thesis, that unified cross-level orchestration is both necessary and would produce meaningful gains, is argued by process of elimination rather than demonstrated empirically.

CPS and ITTS are mathematically defined but never tested, so we have no idea whether they actually discriminate meaningfully between systems.

**Requested Changes:**

Critical for acceptance:

Tables 2 through 4 mix numbers from different models, hardware setups, and baselines. The footnotes help, but the tables still read as if rows are comparable. Either normalize to a shared baseline/model family, or add a visible caveat inside the table itself (not buried in footnotes) making clear that cross-row comparisons are not valid.

Figure 3 is placed prominently and framed as an important illustration of hybrid method behavior, but it is entirely conceptual. Either ground it with real data from at least one hybrid system showing acceptance rates across varying task entropy, or move it to a less prominent position and label it more clearly as speculative.

The Orchestration Gap argument needs more substance. Arguing that each level has ceilings does not establish that those ceilings are binding at the same time in realistic workloads. Provide at least one concrete worked example or case study where a real deployment scenario demonstrably suffers from lack of cross-level coordination

Apply CPS and ITTS retrospectively to a handful of surveyed methods using their reported numbers, even if the calculation is approximate. Without any empirical application, these metrics read as ideas rather than contributions.

---

> ### Author Response · Authors · 2026-06-07
> **Response to Reviewer gsFh**
>
> Thank you for this detailed and constructive critique. We respond to each point below.
>
> &nbsp;
>
> **1.** We agree that footnotes alone do not adequately signal heterogeneity. We will add a visible caveat directly within each table header making clear that rows span different model families, hardware configurations, and baseline types, and that cross-row numerical comparison is not valid. We note the tables already apply substantial exclusion criteria, as methods lacking consistent baselines (e.g., Prompt Lookup Decoding, DRAGIN, ConFu, ThoughtMani) were omitted for this reason, but this is not visible at the point of reading. Normalization to a shared baseline is not feasible given the remaining hardware diversity, but the header caveat is a straightforward fix we will implement.
>
> &nbsp;
>
> **2.** We accept that Figure 3's placement and framing overstate the empirical status of the curves. We considered whether the figure could be grounded with real acceptance-rate data from the surveyed papers; however, no surveyed paper reports acceptance rate as a continuous function of task entropy across a shared axis, as results are reported at fixed benchmark points rather than across a varying entropy spectrum, making direct grounding infeasible without new experiments, which falls outside the scope of a survey contribution. We will reposition the figure and strengthen the caption to make unambiguously clear that the curves are a qualitative illustration of trends observed across the surveyed literature and do not represent empirical measurements. We believe this is the more appropriate resolution for a survey paper.
>
> &nbsp;
>
> **3.** We agree that cataloguing per-level ceilings does not establish that they are simultaneously binding in practice. We will add a short worked example in Section 6.1 tracing a plausible production scenario of a mixed-complexity query batch under memory-constrained serving conditions through all four hybrid levels using explicit illustrative numerical assumptions (e.g., a fixed latency budget, a representative memory ceiling) to show where each level's limits are reached and why no single level can compensate for the others. We want to be clear that this example relies on illustrative assumptions rather than experimental measurement; however, we believe a concrete numerical scenario is more useful to the reader than a purely qualitative narrative and does not require new experiments. The ceiling concepts are already established in the synthesis sections of 4.1 through 4.4 (Heuristic Coupling, Fixed Model Horizon, Auxiliary Model Overhead, Persistent Verification Scaling, Semantic Opacity, Configuration Brittleness) and the example will cite these directly rather than re-derive them.
>
> &nbsp;
>
> **4.** As shown in Equation 3 of the paper, the CPS ratio between a hybrid system and a baseline simplifies to $A \times S$ (accuracy ratio times wall-clock speedup) when both systems are evaluated on the same fixed benchmark; we will include the full derivation of this simplification in Section 5.1.5 of the revision. We apply this to two methods from Table 3:
>
> **SpecReason (Pan et al., 2025):** Speedup $1.4\times$ to $3.0\times$, accuracy improvement $+0.4\%$ to $+9.0\%$ on AIME, MATH-500, GPQA. Treating percentage point increases as approximate relative multipliers gives a CPS ratio of $1.4 \times 1.004 \approx 1.41\times$ to $3.0 \times 1.09 \approx 3.27\times$ over vanilla AR.
>
> **Judge Decoding (Bachmann et al., 2024):** Speedup up to $9\times$, quality preserved on GSM8K, MT-Bench, ARC, MMLU, giving $A \approx 1.0$ and CPS ratio $\approx 9\times$.
>
> We will include these in Section 5.1.5 with explicit caveats: ratios are relative not absolute; figures may reflect slightly different hardware configurations or prompt distributions; and accuracy is treated as binary, which the paper already acknowledges as a limitation of CPS for open-ended tasks. Critically, the fact that even these retrospective calculations require explicit range-reporting and approximation assumptions is itself direct evidence for the standardised joint reporting our paper advocates, and the examples serve as both an illustrative demonstration and as a diagnosis of the current evaluation gap.
>
> &nbsp;
>
> **Note on section numbering**: In our revised manuscript we restructured the paper so that Evaluation Metrics and Datasets (formerly Section 6) now precedes Research Gaps and Future Directions (formerly Section 5). All section references in this response reflect the numbering in the revised manuscript.